# The transcript cleavage factor paralogue TFS4 is a potent RNA polymerase inhibitor

Thomas Fouqueau [1], Fabian Blombach [1], Ross Hartman[2], Alan C.M. Cheung [1], Mark J. Young[2,3] & Finn Werner[1]

TFIIS-like transcript cleavage factors enhance the processivity and fidelity of archaeal and eukaryotic RNA polymerases. *Sulfolobus solfataricus* TFS1 functions as a bona fide cleavage factor, while the paralogous TFS4 evolved into a potent RNA polymerase inhibitor. TFS4 destabilises the TBP–TFB–RNAP pre-initiation complex and inhibits transcription initiation and elongation. All inhibitory activities are dependent on three lysine residues at the tip of the C-terminal zinc ribbon of TFS4; the inhibition likely involves an allosteric component and is mitigated by the basal transcription factor TFEα/β. A chimeric variant of yeast TFIIS and TFS4 inhibits RNAPII transcription, suggesting that the molecular basis of inhibition is conserved between archaea and eukaryotes. TFS4 expression in *S. solfataricus* is induced in response to infection with the *Sulfolobus* turreted icosahedral virus. Our results reveal a compelling functional diversification of cleavage factors in archaea, and provide novel insights into transcription inhibition in the context of the host–virus relationship.

[1] Institute of Structural & Molecular Biology, Division of Biosciences, University College London, London WC1E 6BT, UK. [2] Department of Microbiology, Montana State University, 173520, Bozeman, MT MT 59717, USA. [3] Department of Plant Sciences, Montana State University, 173150, Bozeman, MT MT 59717, USA. Correspondence and requests for materials should be addressed to F.W. (email: f.werner@ucl.ac.uk)

Evolutionary-related multisubunit RNA polymerases (RNAPs) carry out DNA-dependent transcription in all domains of life, bacteria, archaea and eukarya[1]. The shape of the universally conserved RNAP core resembles a crab claw with a DNA-binding channel facilitating interactions with the DNA template and the secondary channel, also called the NTP entry channel (termed funnel and pore in eukaryotic RNAPs)[2]. The secondary channel serves as a binding site for regulatory factors including transcript cleavage factors, and as exit channel for the RNA 3′ end during backtracking[3–7]. In the transcription elongation phase, RNAP frequently pauses in response to pause signals in the template sequence, due to obstructions including DNA-bound proteins or DNA lesions. Following pausing, RNAP moves in a retrograde direction along the template in a process called backtracking[8, 9]. Backtracking causes the 3′ end of the nascent RNA to be displaced from the RNA/DNA hybrid and extruded through the RNAP secondary channel, rendering the transcription elongation complexes (TEC) inert due to the absence of a RNA 3′ hydroxy group in the active site[4, 10]. Transcript cleavage factors such as Gre in bacteria and TFIIS/TFS in eukaryotes and archaea, respectively, assist arrested TECs by enhancing the RNA cleavage activity intrinsic to RNAP[4, 11–13]. These cleavage events release small RNA fragments and result in

a new RNA 3′ terminus in the RNAP active site, which is able to resume transcription elongation. Backtracking is also induced by the misincorporation of nucleotides. Transcript cleavage factors therefore not only facilitate high TEC processivity but they also increase the fidelity of gene expression[7]. While Gre and TFS/TFIIS factors are not related on the sequence or structural level, they carry out analogous functions in bacteria and archaea/eukaryotes and thus provide a compelling case of convergent evolution that highlights the strong need to provide a solution for conflicts arising from arrested TECs[14]. Three basal factors form the archaeal preinitiation complex (PIC) and facilitate transcription initiation in archaea: the TATA box-binding protein TBP, transcription factor B (TFB) and E (TFE), which are homologous to the eukaryotic RNAPII factors TBP, TFIIB and TFIIE, respectively[1, 15–17]. Besides its role in transcription elongation, TFIIS is also incorporated into the RNAPII PIC, although its role during transcription initiation is only poorly understood[18–20]. Whether TFS can be similarly incorporated into the archaeal PIC remains unclear.

TFIIS-like factors are present in all eukaryotic, archaeal and some megavirus RNAP systems[21]. TFIIS and the RNAPII subunit RPB9 are paralogous (Fig. 1a), but RPB9 has lost its ability to act as a transcript cleavage factor[22]. In contrast, the RPB9 paralogues

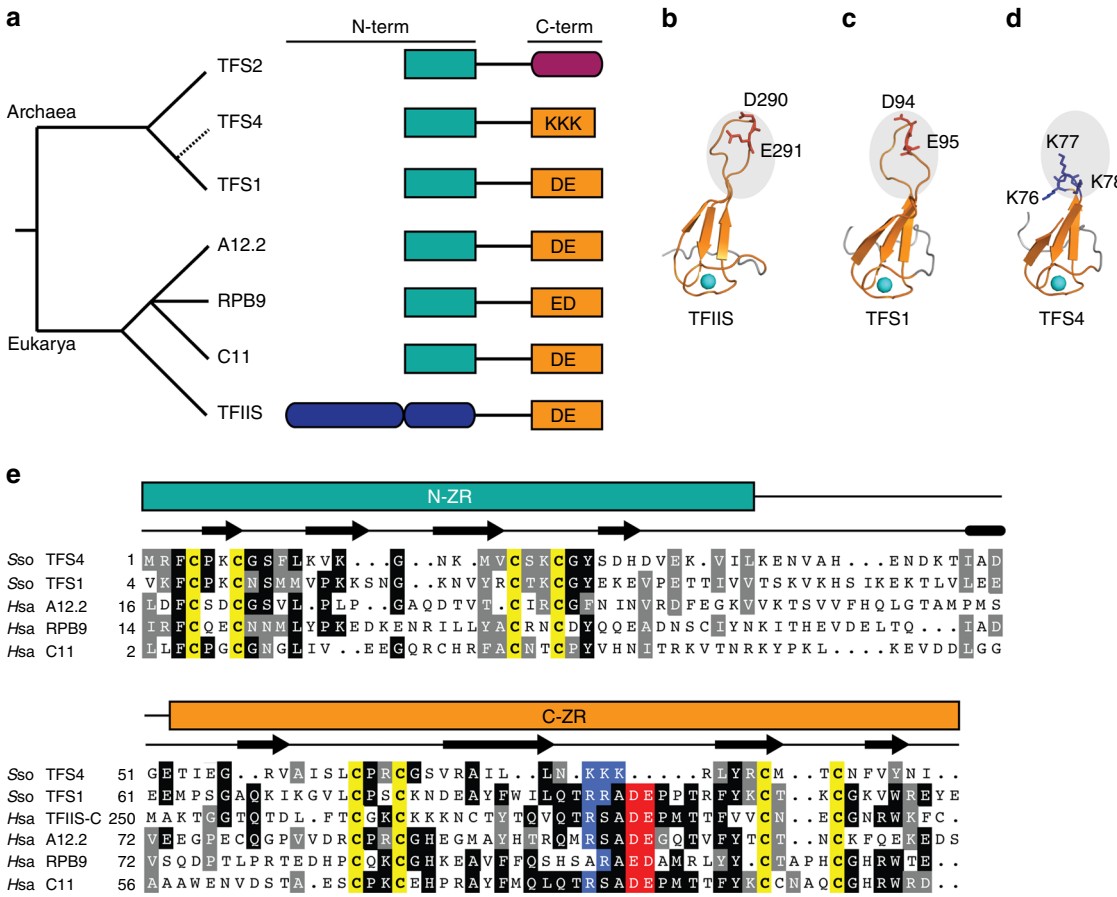

**Fig. 1** Evolutionary conservation of TFIIS-like proteins. **a** Phylogenetic distribution and domain conservation of TFS-related proteins in archaea and eukaryotes. Gene duplication events specific to the Sulfolobales order are shown as dashed lines. Zinc ribbon and α-helical domains as symbolized by rectangles and ellipses, respectively. **b–d** Structural homology between yeast TFIIS (**b**), Sso TFS1 (**c**) and Sso TFS4 C-ZR (**d**). Structural models of the C-ZR domains of TFS1 (C-score: 0.63; TM-score: 0.80±0.09) and TFS4 (C-score: −0.03; TM-score: 0.71±0.12) were prepared using PHYRE 2[56]. The two carboxylate residues (TFIIS and TFS1) and the lysine residues (TFS4) are highlighted as stick representation in red and blue, respectively. Zn²⁺ ions are shown as cyan spheres. **e** Sequence alignment of of *Sulfolobus solfataricus* (*Sso*) TFS4 (AAK42105.1), Sso TFS1 (AAK40629.1), *Homo sapiens* (*Hsa*) RPB9 (P36954.1), *Hsa* A12.2 (Q9P1U0.1), Hsa C11 (AAD31424.1), *Hsa* TFIIS C-ZR (AAH72460.1). Secondary structure prediction of *Sso* TFS1 is shown above the alignments

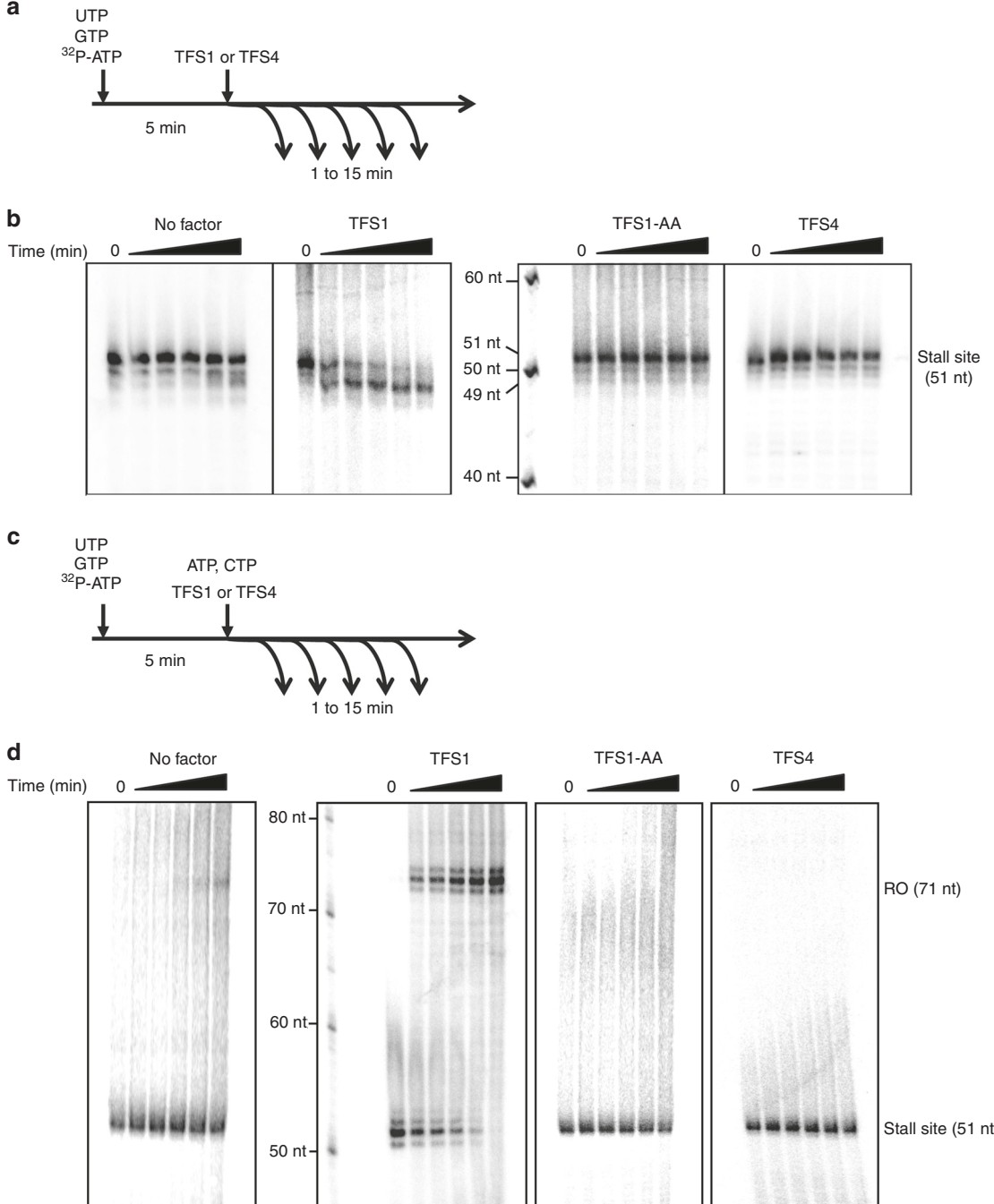

**Fig. 2** TFS1 but not TFS4 stimulates RNAP transcript cleavage activity. **a** Outline of transcript cleavage assays. The reactions containing A*/G/UTP mix and the SSV1-T6 promoter fused to a 50 nt C-less cassette were incubated at 75 °C. After 5 min, TFS1 or TFS4 was added to the reaction. Samples were collected at various time intervals (0, 1, 2, 5, 10 and 15 min). **b** Denaturing PAGE analysis of transcript cleavage products in the presence or absence of TFS1, TFS1-AA or TFS4. **c** Outline of reactivation assays. The reactions were performed with A*/G/UTP mix using SSV1-T6 promoter fused to 50 nt C-less cassette. After 5 min at 75 °C, TFS1 or TFS4 was added and the reaction was chased with CTP and an excess of ATP. Samples were collected at various time intervals (0, 1, 2, 5, 10 and 15 min). **d** Denaturing PAGE analysis of reactivation products in the presence or absence of TFS1, TFS1-AA or TFS4

A12.2 and C11 which are stably incorporated into RNAPI and III, respectively, function as inbuilt transcript cleavage factors[21, 23]. Like A12.2, RPB9 and C11, archaeal TFS consists of two Cys$_4$ zinc ribbon domains called N-ZR and C-ZR[9, 21]. The archaeal TFS factors are closely related to RPB9-like RNAP subunits on the sequence level, but have been shown to reversibly associate with RNAP and stimulate transcript cleavage similar to TFIIS[9, 21]. The structures of the bacterial RNAP-GreB and yeast RNAPII–TFIIS complexes reveal that both factors insert two acidic residues (Asp-

Glu) located at the tip of an elongated domain through the secondary channel into the RNAP active site[3, 24–26]. This stabilises the chelation of the second RNAP active site magnesium ion (Mg-B), which activates a water molecule and facilitates the nucleophilic attack on the phosphodiester backbone of the RNA resulting in cleavage[26, 27]. In effect, TFIIS contributes to a 'composite' active site that carries out endonucleolytic RNA cleavage.

Here, we describe the discovery of a fascinating multiplication of *tfs*-like genes in the crenarchaeon *S. solfataricus* (Sso) and

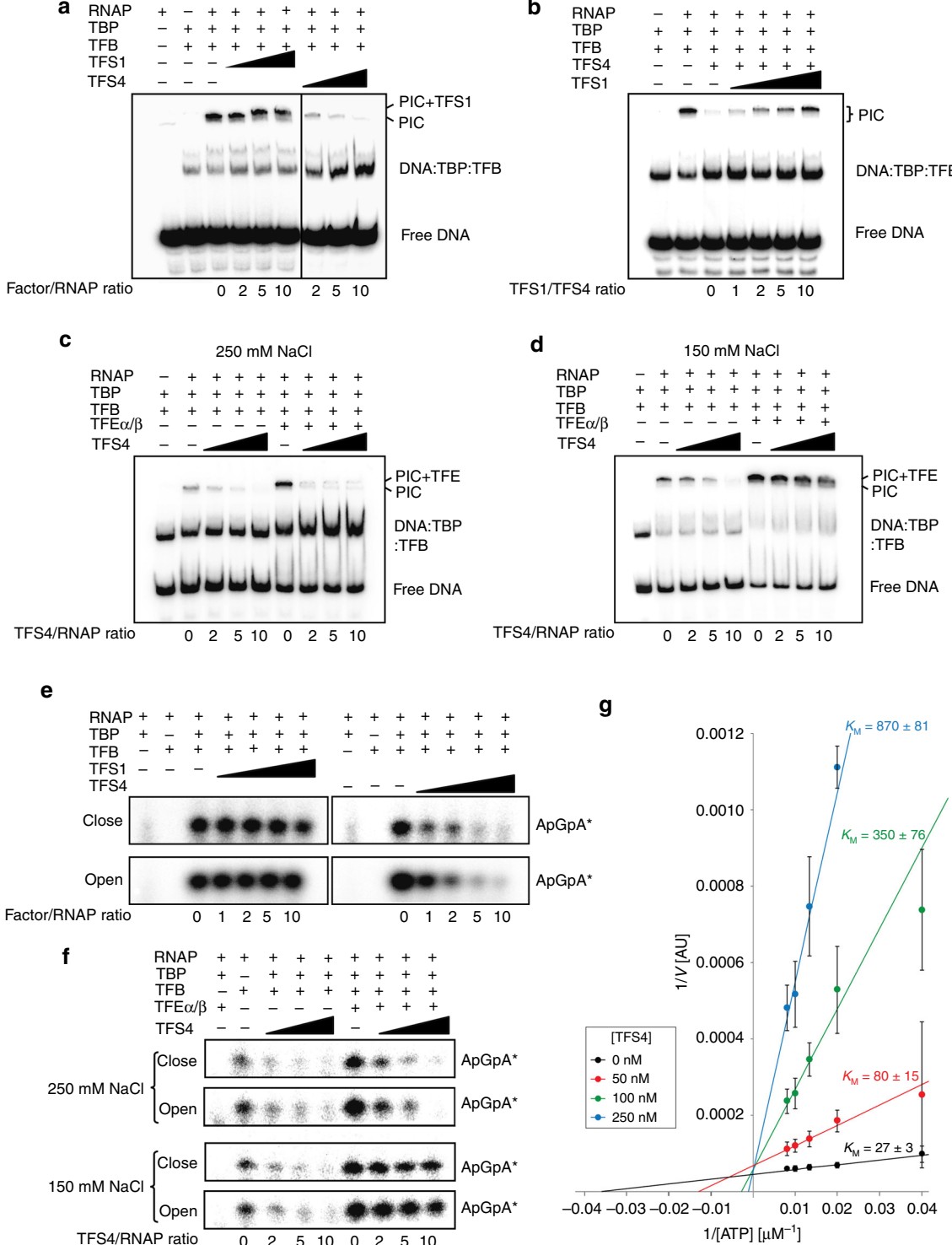

**Fig. 3** TFS4 inhibits pre-initiation complex formation and abortive transcription. **a** Electrophoretic mobility shift assays (EMSAs) with the minimal archaeal PIC complex (SSV1-T6 promoter DNA, TBP, TFB, RNAP) in the presence of TFS1 or TFS4. **b** Increasing amount of TFS1 can rescue the formation of the minimal archaeal PIC pre-incubated with 100 nM TFS4. **c**, **d** TFEα/β mitigates the inhibition of PIC formation by TFS4 under low salt (150 mM) (**d**) but not under high salt concentration (250 mM) (**c**). **e** Abortive transcription assays measuring the addition of [α32P]-ATP to the dinucleotide primer ApG on homoduplex (close) and heteroduplex (open) SSV1-T6 promoter template in the presence or absence of TFS1 or TFS4. **f** TFEα/β mitigates the inhibitory activity of TFS4 on addition of [α32P]-ATP to the dinucleotide primer ApG under low salt (150 mM) (lower panel) but not under high salt concentration (250 mM) (higher panel). **g** Double reciprocal plots showing abortive initiation versus substrate NTP concentration in the absence or presence of TFS4. Error bars represent standard deviation from three technical repeats

present a functional and structural characterisation of two proteins, TFS1 and TFS4, that display opposite stimulatory and inhibitory effects on transcription. While TFS1 has the canonical transcript cleavage activity and enhances elongation, TFS4 has evolved into a potent inhibitor of RNAP. TFS4 inhibits (i) PIC formation, (ii) abortive initiation, (iii) promoter-directed transcription as well as (iv) transcription elongation. Domain swapping and mutagenesis experiments have identified three lysine residues located at the tip of the C-ZR domain that are required for the inhibitory activities of TFS4. In line with these potentially detrimental properties, overexpression of a TFS1/TFS4 hybrid carrying the three lysine residues causes growth retardation in *Sulfolobus*. Interestingly, infection of Sso with the *Sulfolobus* turreted icosahedral virus (STIV) induces TFS4 expression, suggesting that TFS4 plays a role in the antiviral host response and defence.

## Results

**Identification of TFS homologues in *S. solfataricus*.** We analysed the phylogenetic distribution and diversity of transcript cleavage factor homologues in archaea. Our results show that the multiplication of *tfs*-like genes occurred independently multiple times in the Cren-, Loki-, Thor- and Heimdallarchaea, as well as in the euryarchaeal Methanobacteria and Halobacteria (Supplementary Table 1). We focused on the biochemically tractable crenarchaeal model organism Sso and identified four TFS paralogues that are henceforth referred to as TFS1–4 (Fig. 1). The amino acid sequence alignment and the structural homology model of TFS1 (*sso0291*) suggest that this paralogue is the canonical TFIIS-related factor homologous to the functionally characterised TFS variant from *Methanothermococcus thermolithotrophicus*[9]. TFS1 contains the catalytic Asp and Glu residues (D94 and E95) at the tip of the C-ZR that are critical for transcript cleavage activity (highlighted in red in Fig. 1b, c, e)[9]. TFS2 (*sso0605*) is conserved in all Crenarchaeota but lacks the C-ZR, which is substituted by a glutamic acid-rich domain predicted to adopt an extended α-helical structure (Supplementary Fig. 1a). TFS3 (*sso5576*) is the most divergent TFS paralogue; it has only two out of eight cysteine residues required for the coordination of the two canonical zinc ions in the two ZR domains, respectively (Supplementary Fig. 1b). Finally, TFS4 (*sso9221*) shares the highest sequence identity with TFS1 (34%) among all TFS paralogues, including the eight cysteine residues to form the two ZR domains with one important exception: the two catalytic acidic residues are missing and the shorter C-ZR hairpin projects three lysine residues in TFS4 (highlighted in blue in Fig. 1d, e). This suggests that TFS4 could function as factor-regulating RNAP through the secondary channel similar to TFS1. However, due to the C-ZR lacking the catalytic residues, TFS4 is predicted to be unable to promote transcript cleavage and rather carries out a different function. TFS4 is conserved in most but not all Sulfolobales species and all TFS4 homologues include the lysine residue motif (Supplementary Fig. 1c, d). In order to explore a possible functional divergence of transcript cleavage factors TFS1 and TFS4, we carried out a multidisciplinary functional characterisation of TFS1 and TFS4 using in vitro and in vivo approaches.

**TFS1 but not TFS4 stimulates transcript cleavage by RNAP.** The *M. thermolithotrophicus* TFS stimulates the transcript cleavage activity of its cognate RNAP akin to its eukaryotic TFIIS homologue[7–9]. In order to test whether TFS1 and TFS4 have this activity, we developed a transcript cleavage assay using transcription templates containing a 50-nt C-less cassette (Fig. 2a; Supplementary Fig. 2b). Initiation of transcription by the addition

of RNAP and the two essential basal transcription factors TBP and TFB in the presence of ATP, GTP and UTP resulted in TECs stalling after synthesising a 51-nt transcript. This transcript is generated by the misincorporation of a single nucleotide at the 3′ end of the 50-nt C-less transcript, a phenomenon observed in archaeal RNAP[8] as well as eukaryotic RNAPII[28]. Without exogenous cleavage factors, only small amounts of 49-nt cleavage products were generated even after extended incubation times (Fig. 2b, 15 min). In contrast, the addition of wild-type TFS1 induced a rapid cleavage generating predominantly a 49-nt RNA species already after 1 min. This cleavage activity is specific since an active site mutant of TFS1 containing alanine substitutions at residues D94 and E95 (TFS1-AA) was inert during 15 min of incubation. The addition of TFS4 did not induce cleavage and the transcript pattern and was indistinguishable from that obtained with TFS1-AA (Fig. 2b). In order to test whether the stalled TECs were able to resume transcription and produce a 71-nt run-off transcript, we chased the reaction by adding the missing fourth nucleotide, CTP (Fig. 2c). Without transcript cleavage factor, only a negligible fraction of the 51-nt transcript (containing the misincorporated 3′-end nucleotide) could be extended, in agreement with the cleavage assay results discussed above. In contrast, the addition of TFS1 facilitated fast and complete resumption of transcription (Fig. 2d), while neither the TFS1-AA mutant nor TFS4 were able to rescue stalled TECs (Fig. 2d). These results demonstrate that TFS1 but not TFS4 stimulates transcript cleavage.

**TFS4 inhibits PIC formation and abortive initiation.** The first step of transcription initiation is the TBP/TFB-dependent recruitment of RNAP to the promoter. We carried out electrophoretic mobility shift assays (EMSAs) using the strong SSV1-T6 promoter to assess the impact of TFS1 and TFS4 on RNAP recruitment and PIC stability[29]. The two initiation factors TBP and TFB are necessary and sufficient to enable RNAP recruitment and the formation of stable PICs on the T6 promoter (Fig. 3a). The addition of TFS1 led to the formation of a lower mobility complex due to its incorporation into the PIC, but otherwise had little effect on the stability of the complex. In contrast, the addition of TFS4 led to the disappearance of the band corresponding to the PIC and a concomitant increase in the ternary DNA–TBP–TFB complex in a dose-dependent fashion, which implies that TFS4 leads to the dissociation of the PIC (Fig. 3a). Considering the high degree of conservation between TFS1 and TFS4, it is likely that they share the same binding site on RNAP. In agreement with this hypothesis, the addition of TFS1 counteracted the inhibitory effect of TFS4 on PIC formation in a dose-dependent manner (Fig. 3b). Since a tenfold excess of TFS1 over TFS4 barely restores the PIC signal in the EMSA, the apparent affinity of TFS4 for the PIC appears to be higher than that of TFS1. Since PIC formation precedes catalysis, the destabilising effect of TFS4 likely has an allosteric component.

TFEα/β, the third basal transcription factor in archaea, is not strictly required but stimulates transcription initiation in several ways. TFEα/β assists open-complex formation and stabilises the PIC through interactions with the DNA non-template strand (NTS) and by inducing conformational changes within RNAP[29, 30]. The stimulatory activity of TFEα/β varies with the ionic strength of the binding buffer at low (150 mM) and high (250 mM) NaCl concentrations (Fig. 3c, d)[29, 31]. To test if TFEα/β can counteract the PIC-destabilising activity of TFS4, we carried out EMSAs in the presence or absence of TFEα/β at high or low ionic strength. Under low-salt conditions, TFEα/β largely prevented the TFS4-induced PIC dissociation, while under high-salt conditions, the PIC was rendered sensitive to TFS4 (Fig. 3c, d). In line with

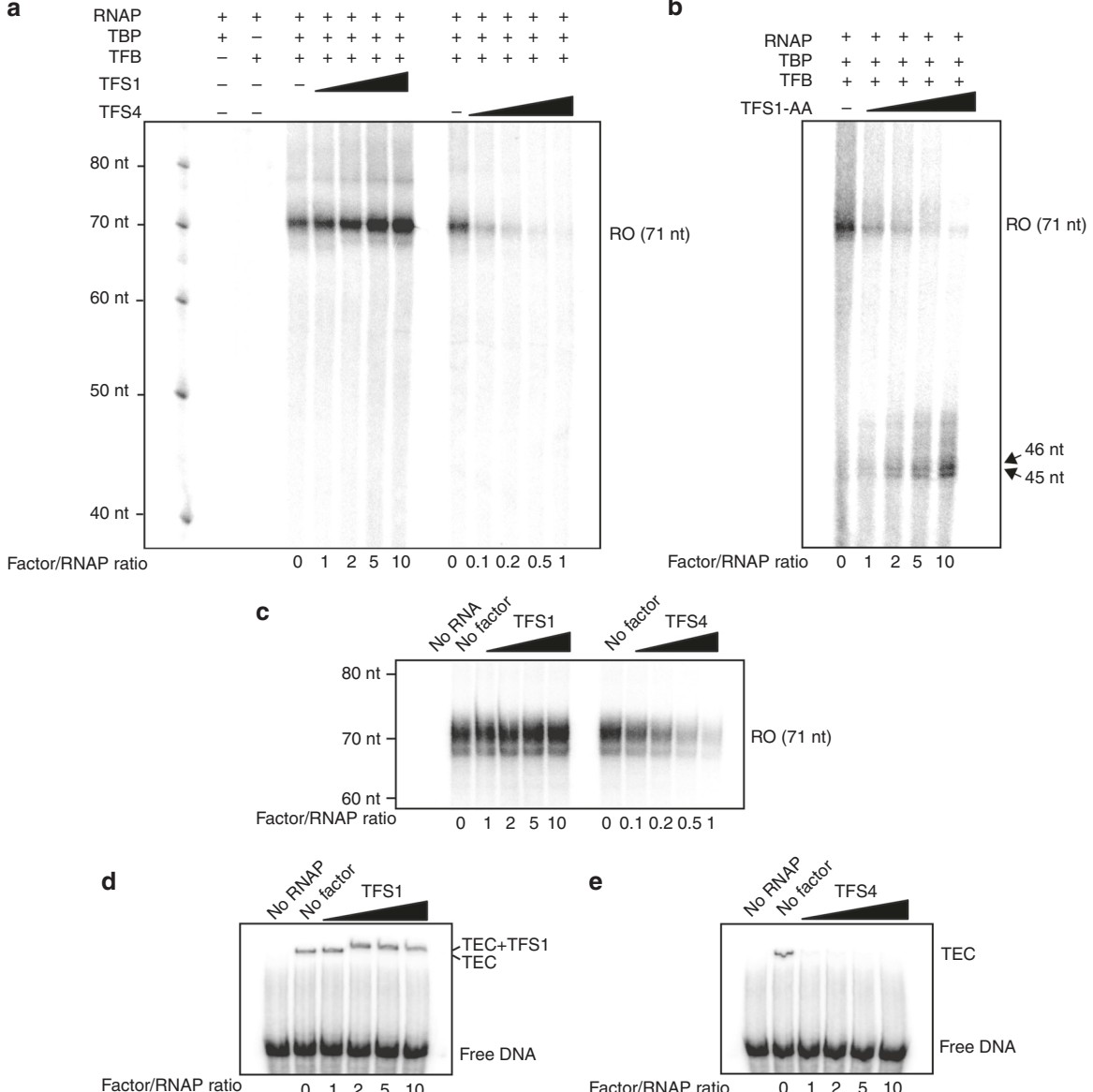

**Fig. 4** TFS4 inhibits promoter-directed transcription initiation, and transcription elongation. **a**, **b** Promoter-directed transcription assays directed by the SSV1-T6 promoter in the presence of TFS1, TFS4 (**a**) or TFS1-AA (**b**). The addition of TFS1 stimulates productive transcription whereas TFS4 inhibits it. The TFS1-AA active site mutant reduces transcription processivity. **c** Transcription elongation assays using a synthetic scaffold template in the presence or absence of TFS1 or TFS4. **d**, **e** EMSA assays using the archaeal TEC consisting of RNAP, radioactively labelled DNA template, and a 14 nucleotide RNA primer in the presence of TFS1 or TFS4. While TFS1 binding does not affect the stability of the TEC, the addition of TFS4 leads to its dissociation

these results, permanganate foot-printing data on DNA opening during open-complex formation suggested that TFS4 inhibits open-complex formation, whereas PICs stabilised by TFEα/β appeared to be insensitive to TFS4 inhibition (Supplementary Fig. 3). No alterations of DNA opening were observed in the presence of TFS1. In order to determine whether TFEα/β and TFS1 or TFS4 could bind to RNAP in the PIC simultaneously, we carried out EMSA supershift experiments. Both TFS1 and TFS4 lead to a subtle but highly reproducible supershift of TFEα/β-containing PICs (Supplementary Fig. 4).

To examine the influence of TFS1 and TFS4 on the formation of the first phosphodiester bond, we used abortive initiation assays that measure the addition of a single NTP to a dinucleotide primer substrate[29]. In this assay, we also compared 'closed' and 'open' templates that are complementary throughout or contain a 4-bp region of non-complementarity upstream of the

transcription start site (TSS) and mimic the closed and open PIC, respectively[29] (Supplementary Fig. 2a). Synthesis of the trinucleotide product was strictly dependent on TBP and TFB, the dinucleotide primer and complementary NTP substrate (Supplementary Fig. 5). The addition of TFS1 had no discernible effect on abortive initiation on either closed or open promoter templates (Fig. 3e, left panel). In contrast, the addition of TFS4 had a strong inhibitory effect on abortive initiation, independent of the closed or open state of the promoter (Fig. 3e, right panel). In agreement with the EMSA results, the addition of TFEα/β in low but not high ionic strength counteracted TFS4 inhibition (Fig. 3f).

To further elaborate on the underlying mechanism of inhibition, we tested whether the TFS4 inhibition varied with NTP concentrations. We carried out abortive initiation assays at a range of NTP substrate- and TFS4 concentrations, and the results are illustrated as a double reciprocal plot in Fig. 3g (primary data

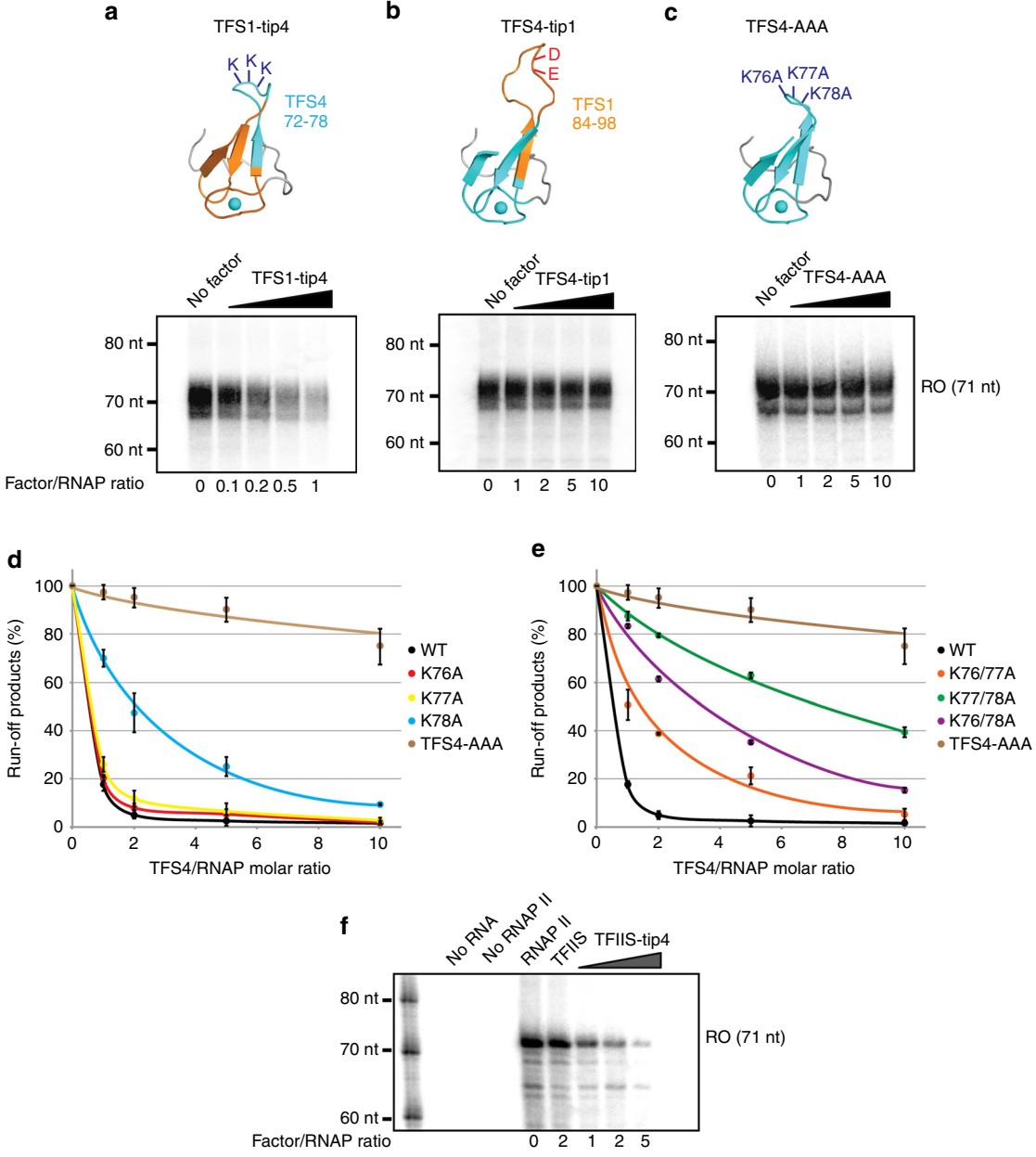

**Fig. 5** Functional activities of chimeric TFS1/TFS4 hybrid- and alanine substitution mutants. **a–c** Structural models of TFS1-tip4, TFS4-tip1 and TFS4-AAA mutants and their inhibitory activity in transcription elongation assays (panels below the structural models). **d, e** Quantified run-off transcript signals from transcription elongation assays carried out in the presence of TFS4 variants with single (**d**) and double alanine substitution (**e**) of the three lysine residues in the C-ZR. Error bars represent standard deviation from three technical repeats. **f** Transcription elongation assays with yeast RNAP II and its response to yeast TFIIS or TFIIS-tip4

in Supplementary Fig. 6a, b). The results show that TFS4 acts as a competitive inhibitor interfering with NTP binding. At 250 nM TFS4 (fivefold excess over RNAP), the apparent $K_m$ for NTP is increased ~30-fold from $27 \pm 3\,\mu M$ to $870 \pm 81\,\mu M$.

**TFS4 inhibits productive initiation and elongation.** Following abortive initiation, the RNAP escapes the promoter and enters the productive transcription elongation phase. To investigate the effect of TFS1 and TFS4 during this stage, we carried out promoter-dependent run-off transcription assays. SsoRNAP synthesises a 71-nt run-off transcript from the SSV-T6 promoter in this assay strictly dependent on TBP and TFB (Fig. 4a; Supplementary Fig. 2c). The addition of TFS1 stimulates

transcription at ~2.5-fold (Fig. 4a) in agreement with its predicted function enhancing transcription processivity[7, 9, 17]. Interestingly the TFS1-AA mutant leads to the accumulation of partial 45/46-nt transcripts (Fig. 4b). The addition of TFS4 efficiently inhibited the synthesis of the 71-nt run-off transcript without leading to the accumulation of partial transcripts (Fig. 4a). The inhibition of catalysis would not result in accumulation of shorter products if the NTP-binding constant for the first round of nucleotide addition during initiation is much higher than the average NTP-binding constant during elongation. This disparity in NTP-binding affinities is typical for multisubunit RNA polymerases because the first round of nucleotide addition requires two NTPs instead of one, and the position of the template strand is not stabilised by an RNA–DNA hybrid. TFS4 inhibition was very

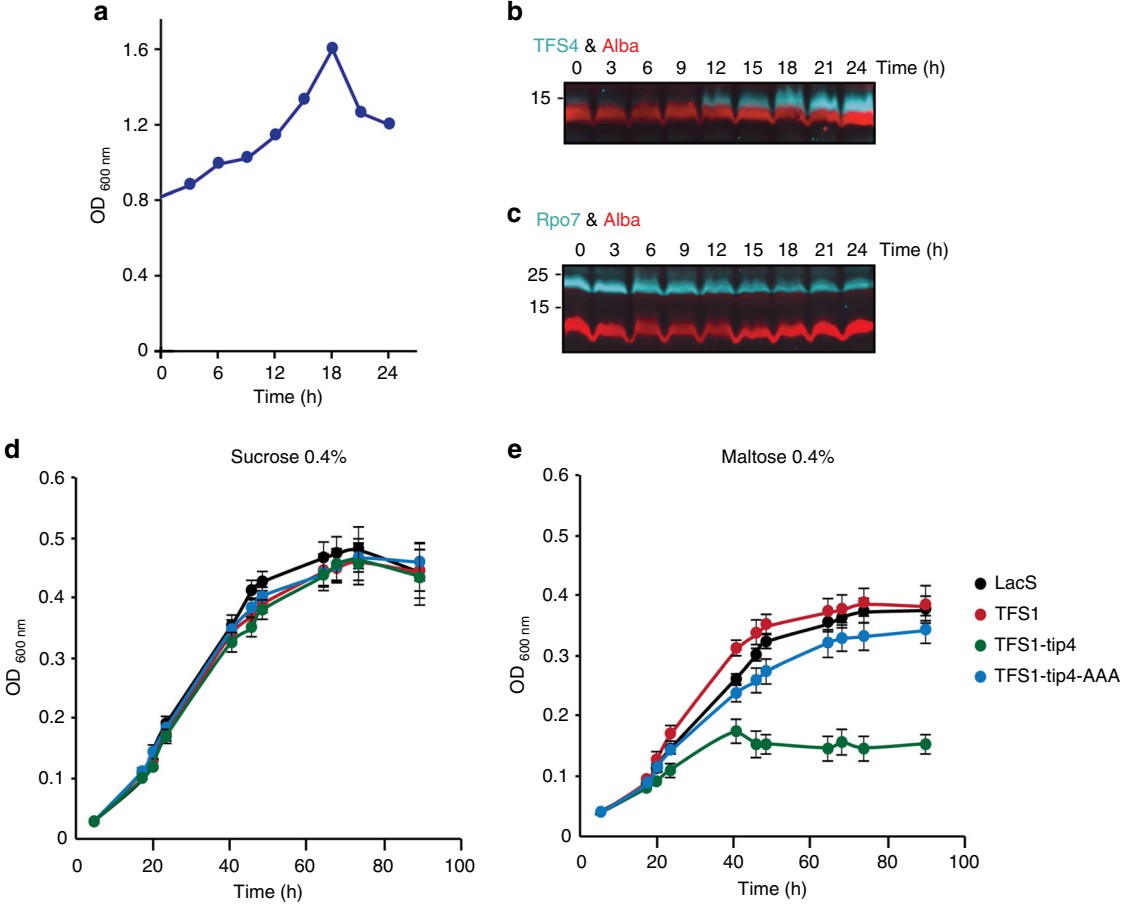

**Fig. 6** STIV virus infection induces TFS4 expression levels and growth retardation. **a** Infection time course experiment. The growth curve shows the increase of the optical density of *S. solfataricus* (strain 2-2-12) following infection by the STIV virus ($t = 0$). **b, c** Multiplex Western blot using polyclonal antibodies raised against TFS4 (**b**), RNAP subunits Rpo4/7 (**c**) and Alba (loading control) as a function of time following infection by STIV. Immunodetection was performed on two biological replicates. **d** Growth curves of *S. acidocaldarius* MW001 harbouring expression vectors encoding TFS1, TFS1-tip4, TFS1-tip4-AAA and LacS (control) under non-inducing or inducing **e** conditions using sucrose or maltose containing media, respectively. Error bars represent standard deviation from three technical repeats

efficient since it was able to fully inhibit transcription at equis-toichiometric TFS4:RNAP concentrations about 10× lower than the concentration required to inhibit PIC formation. The TFEα/β protection against TFS4 inhibition was less pronounced in the promoter-directed run-off experiments compared to PIC forma-tion and abortive initiation (Supplementary Fig. 7a, b). Taken together, these results indicate that TFS4 also targets the elon-gation phase of the transcription cycle. In order to directly test the effect of TFS1 and TFS4 during elongation in a promoter- and TBP/TFB/TFEα/β-independent manner, we carried out tran-scription elongation assays using a synthetic scaffold template[32] (Supplementary Fig. 2d). While TFS1 stimulated elongation at high TFS1-RNAP ratios (tenfold), TFS4 inhibited RNAP at much lower (equistoichiometric) TFS4-RNAP ratios (Fig. 4c). Con-sidering that the RNAP-binding sites of TFS1 and -4 likely are identical or at least overlap, we wanted to test whether TFS1 was able to counteract TFS4 inhibition in transcription elongation assays. Under our experimental conditions, a ~twofold excess of TFS1 over TFS4 was required to achieve 50% relief of inhibition, which indicates that TFS4 has a higher apparent affinity for RNAP in the TEC compared to TFS1 (Supplementary Fig. 8a, b).

Since our previous experiments showed that at least part of the TFS4 inhibition was due to a destabilisation of the protein–nucleic acid interactions within the PIC, we tested whether TFS1 and TFS4 could destabilise TECs. RNAP formed

heparin-stable TECs on the synthetic elongation scaffolds (Fig. 4e, d). The addition of TFS1 did not destabilise the TEC but led to a decrease in TEC electrophoretic mobility indicative of TFS1 incorporation (Fig. 4d). In contrast, the addition of TFS4 led to the complete disappearance of the TEC signal (Fig. 4e).

**TFS4 inhibition is facilitated by three lysine residues**. The competition between TFS1 and -4 for RNAP interactions in EMSA experiments suggests that the binding sites of TFS1 and TFS4 on RNAP are partially overlapping if not identical. The catalytic residues at the tip of the C-ZR domain are the key difference between TFS1 and -4; while TFS1 has two acidic residues (D94 and E95), TFS4 has three lysine residues (K76, K77 and K78) (Fig. 1c, d). To investigate whether the lysine-containing TFS4 C-ZR tip is necessary and sufficient for the inhibition of RNAP, we created two chimeric TFS variants by swapping the tip motif of the C-ZR (including DE or KKK motifs) between TFS1 (residues 84–98) and TFS4 (residues 72–78). The TFS1-tip4 variant consists of TFS1 and the C-ZR tip from TFS4, while TFS4-tip1 corresponds to TFS4 with the C-ZR tip from TFS1 (Fig. 5a, b). Transcription elongation assays demonstrate that the TFS1-tip4 mutant efficiently inhibits tran-scription comparably to TFS4 (Fig. 5a). Likewise, EMSA experi-ments confirm that the TFS1-tip4 variant destabilised the PIC

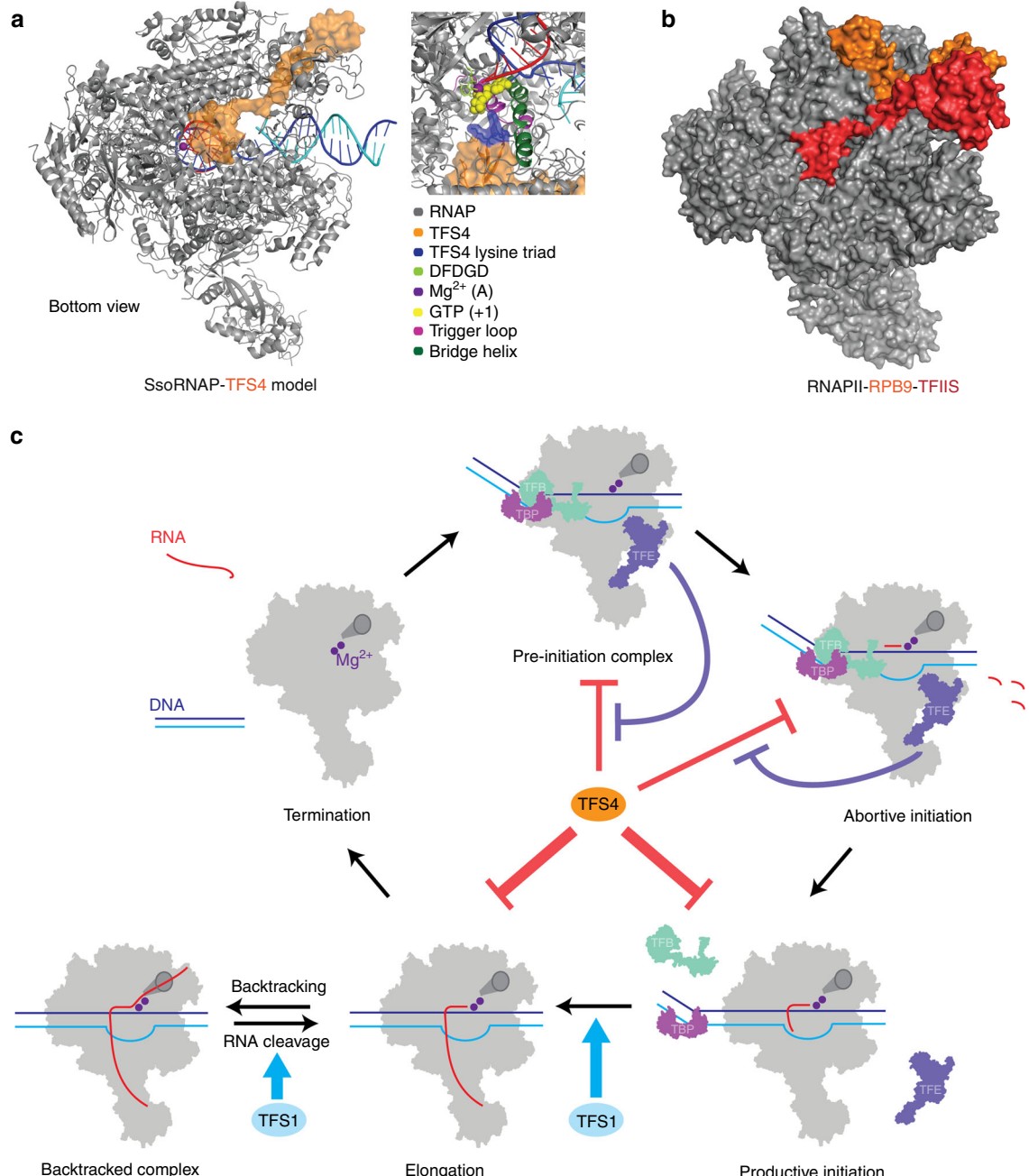

**Fig. 7** Global regulation of SsoRNAP transcription cycle by TFS1 and TFS4. **a** Structural homology model of the Sso RNAP-TFS4 transcription elongation complex in which TFS4 is bound to RNAP akin to the A12 subunit to RNAPI. TFS4 is shown as surface representation in orange. **b** Surface representation of the yeast RNAPII–TFIIS complex highlighting RPB9 and TFIIS in orange and red, respectively. **c** A schematic representation of SsoRNAP transcription cycle. TFS1 stimulates productive transcription and rescues backtracked complexes by stimulating RNA cleavage by RNAP. In contrast, TFS4 inhibits transcription initiation by a two-fold mechanism. TFS4 (i) destabilises pre-initiation complexes (PICs) likely by an allosteric mechanism and (ii) interferes with catalysis by competing with substrate NTP binding. The basal factor TFEα/β stabilises the PIC and counteracts the inhibitory effect of TFS4, possibly by allosteric competition. During the elongation phase of transcription, TFS4 inhibits transcription by destabilising the RNAP–DNA–RNA complex and inhibiting catalysis

and inhibited promoter-directed transcription initiation (Supplementary Fig. 9b, d). To unequivocally prove that the lysine residues are critical for inhibition, we tested the impact of the TFS4-AAA mutant on PIC formation and promoter-directed transcription (Supplementary Fig. 9a, c) and transcription elongation (Fig. 5c). As expected, the TFS4-AAA variant had no inhibitory activities. In order to test whether the triple-lysine motif contributed to the binding of TFS4 to RNAP, we carried out competition experiments between TFS4 and TFS4-AAA in

EMSAs. The addition of TFS4-AAA (up to tenfold excess over TFS4) could only partially rescue the destabilising effect of TFS4 on the PIC, which suggests that the lysine residues do contribute to RNAP binding (Supplementary Fig. 9e). As predicted, the 'inverse chimeric' TFS4-tip1 variant did not inhibit RNAP activity in promoter-directed transcription (Supplementary Fig. 9c) and elongation assays (Fig. 5b), but in turn had attained transcript cleavage activity (Supplementary Fig. 10). In order to determine the significance of the individual lysine residues, we created single

and double alanine substitution mutants. Transcription elongation assays revealed phenotypes of intermediate strength; out of the three single-substitution variants, K78A showed the strongest decrease in transcription inhibition (Fig. 5d), while double mutants generally affected the inhibitory effect more severely (Fig. 5e). In order to test whether the positive charge per se facilitated inhibition, we tested a triple arginine TFS4 variant in EMSA and transcription run-off assay; TFS4-RRR inhibited transcription comparable to TFS4 (Supplementary Fig. 11a, b). Considering the close evolutionary kinship of TFS and TFIIS factors, we tested whether the TFS4 C-ZR tip was able to inhibit *Saccharomyces cerevisiae* RNAPII. Indeed, a chimeric *S. cerevisiae* TFIIS-tip4 variant efficiently inhibited *S. cerevisiae* RNAPII in transcription elongation experiments directly comparable to Sso RNAP (Fig. 5f).

**Virus infection induces TFS4 expression**. Sso cells do not express any detectable amounts of TFS4 transcripts[33] or TFS4 protein during exponential or stationary growth phases (Supplementary Fig. 12a). However, the infection by STIV results in an increase of TFS4 mRNA levels in Sso (strain 2-2-12)[34]. In order to verify the induction of TFS4 expression on the protein level, we used immunodetection in Sso 2-2-12 cell extracts of a STIV post-infection time course (Fig. 6a–c). No TFS4 expression was detectable prior to virus infection at time point zero. The TFS4 Western signal emerges 12 h post infection (p.i.) and increases steadily to the last time point of 24 h p.i. (Fig. 6b). The optical density of the infected culture starts to decrease after 18 h p.i. while TFS4 levels continue to accumulate (Fig. 6a, b). As a control, we monitored the protein levels of the RNAP subunit Rpo7 and the chromatin protein Alba, which remained unchanged during the infection time course. This suggests that the increase in TFS4 levels was specific and not a general of virus infection (Fig. 6c). Since our in vitro analysis suggests that TFS4 is a puissant inhibitor of transcription, we speculated whether transcription inhibition by TFS4 could be the cause of growth retardation. In order to address this question, we studied the effect of TFS4 over-expression on the growth of a closely related *Sulfolobus* species that does not encode a TFS4 homologue, *Sulfolobus acidocaldarius* (Saci). The TFS4 gene (*sso9221*) was cloned under the control of a maltose-inducible promoter[35]. Over-expression of Sso TFS4 in Saci did not yield a stable full-length protein but rather two degradation products (Supplementary Fig. 12b). The fragment sizes suggested they correspond to the N-ZR and C-ZR domains resulting from a proteolytic attack by Saci proteases. In order to overcome this problem, we cloned a chimeric TFS variant encompassing Saci TFS1 with the Sso TFS4 C-ZR tip (Saci TFS1-tip4) analogous to the Sso TFS1-tip4 variant described above. As negative controls for this experiment we used expression vectors encoding wild type Saci TFS1, the triple lysine to alanine substitution variant Saci TFS1-tip4-AAA, and the LacS reporter. During growth in sucrose medium under non-inducing conditions the optical density of all four strains showed similar growth profiles (Fig. 6d). Following transfer to a maltose medium —inducing conditions—the growth of the strain harbouring the Saci TFS1-tip4 expression plasmid was severely impaired, while the Saci TFS1-tip4-AAA mutant showed little effect and the two additional control strains (TFS1 and LacS) continued to grow indistinguishable from the parental strain (Fig. 6e). This suggest a link between transcription inhibition and the growth phenotype. The Saci TFS1-tip4 strain completely recovered when transferred to sucrose media, suggesting that expression of the Saci TFS1-tip4 variant does not lead to cell death but rather to a dormancy-like phenotype (Supplementary Fig. 12c).

## Discussion

Our study provides a fascinating example of the functional diversification of transcript cleavage factor paralogues in the archaea. Whereas TFS1 enhances transcription processivity by stimulating the RNA cleavage activity of RNAP, its paralogue TFS4 has evolved into a potent inhibitor of RNAP. This radical change in functionality only requires a small perturbation of TFS1, a substitution of the canonical C-ZR hairpin, with the two acidic residues, by a shorter hairpin bearing three lysine residues.

Archaeal TFS paralogues have a varied phylogenetic distribution. TFS1 is conserved in all archaea with the sole exception of *Methanopyrus kandleri*[36], TFS2 is conserved among all Crenarchaeota, whereas TFS3 and TFS4 appear to have evolved later and be restricted to the order Sulfolobales within the Crenarchaeota (Supplementary Fig. 1b). The functional key feature of TFS4 —a shortened C-ZR domain with positively charged amino acid residues—is highly conserved among all TFS4 homologues, although the number of residues varies between two and three (Supplementary Fig. 1a). Our motif swapping and mutagenesis analysis of TFS1 and -4 demonstrate that the tip of the C-ZR domain determines whether the Sso TFS factors carry out cleavage or inhibition, that the extent of the inhibition is a function of the number of lysine residues, that the positive charged nature of the residues is the crucial determinant of inhibition, that the charge likely contributes to RNAP binding of the factor, and finally that this mechanism underlying inhibition is conserved between archaeal and eukaryotic TECs. In the crystal structure of the RNAPII–TFIIS complex the TFIIS C-ZR penetrates the secondary channel deep into the RNAPII active site[4, 25]. All our results suggest that TFIIS, TFS1 and TFS4 interact with their cognate RNAP using the same binding site, through the secondary channel, but only TFS4 inhibits RNAP[4, 25]. Based on the high sequence homology between TFS4 and A12/RPB9/C11 including the two ZR domains, we prepared a structural model of the SsoRNAP–TFS4 complex (Fig. 7). In this model, the TFS4 N-ZR domain binds to the funnel region of RNAP and the C-ZR domain is inserted into the secondary channel reaching towards the active site of RNAP in similar manner to the RNAP subunits A12 and C11 in RNAPI and RNAPIII, respectively, distinct from both RPB9 and TFIIS on RNAPII (Fig. 7a, b)[37–39].

How does TFS4 inhibit transcription? The catalytically inactive cleavage factor mutant TFS1-AA impairs processivity in vivo and in vitro in a fashion that is distinct from TFS4 (Fig. 4a). This effect is comparable to the cognate human TFIIS-AA and bacterial GreA mutant variants that also have inhibitory activities and a dominant negative lethal phenotype[40, 41]. The underlying molecular mechanism of TFS4 inhibition has at least two components, (i) a catalytic inhibition manifested by an increased $K_m$ for NTP substrates that can be observed at low TFS4 concentrations, and (ii) a destabilisation of PICs that requires higher TFS4 concentrations. The structural basis of the first mechanism is likely due to the binding of TFS4 in the secondary channel where it could interfere with flexible elements like the trigger loop (TL) and bridge helix (BH) that play a fundamental role in nucleotide selection and the translocation mechanism in multi-subunit RNAPs[8, 42, 43]. Antibiotics and various transcription factors, including Gre and TFIIS modulate RNAP activity by locking the TL in inactive configurations[4, 11, 43, 44]. The second mechanism could be achieved by allosteric changes reminiscent of the Gre homologue Gfh1, which inhibits transcription by inserting a coiled-coil domain into the secondary channel likely akin to TFS4[26, 45]. The engagement of Gfh1 with the bacterial RNAP induces local structural changes in the active site, as well as overall conformational changes in the RNAP. Acidic residues at the tip of the Gfh1 coiled coil domain introduce a kink in the BH and lock the TL in an inactive conformation[5]. Moreover, the two

approximate hemispherical lobes of the RNAP core are undergoing a pivoting movement by ~7°[5]. Large-scale structural changes play an important role in the regulation of RNAP function in other multisubunit RNAPs as well, e.g. the inactive conformation of eukaryotic RNAPI dimers[37]. TFE, like TFIIE interacts with the non-template DNA strand and the RNAP clamp module favoring its open conformation and increasing the stability of the PIC[15, 30, 46]. We propose an allosteric competition between TFE and TFS4, since TFS4 destabilises the PIC, while TFE stabilises the PIC and counteracts the inhibitory activity of TFS4 (Fig. 7c). Starvation and oxidative stress induce dramatic depletion of subunit TFEβ in Sso[29]. It is therefore possible that the inhibitory effect of TFS4 on transcription initiation is more dominant under certain growth conditions.

What is the biological role of TFS4? In their natural habitats, *Sulfolobus* species are under severe attack from viruses as testified by the presence of three different types of CRISPR–Cas systems and a large number of transposons in the Sso genome[47, 48]. Since TFS4 expression is strongly induced in response to virus infection, its likely to play a role in host defence. STIV-infected Sso 2-2-12 cells induce expression of TFS4 but not any of the CRISPR system components[34]. Considering that the over-expression of TFS1-tip4 results in growth inhibition, and that the onset of virally induced growth inhibition follows the increase in TFS4 expression levels, it is tempting to speculate that TFS4 expression contributes to growth inhibition in agreement with its function as global transcription inhibitor. Predation by viruses frequently induces a dormant or quiescent state in infected host cells, which serves as a type of hiatus that can facilitate survival by 'weathering out' the attack and/or by maintaining the virus within the cell for a long time enough to activate an alternative host defense[49]. This 'persistence' is a well characterised response to infections in bacteria and eukaryotes, and is a highly adaptive trait provided that the survivors manage to live once the phage/virus is cleared[50].

In conclusion, we present the first functional characterisation of a novel archaeal TFS/TFIIS-like factor that does not function as a transcript cleavage factor, but rather functions as a global inhibitor of RNAP. Unlike the canonical TFS1, which stimulates transcriptional activity and transcript cleavage of RNAP, TFS4 induces the efficient inhibition of RNAP transcription. Because infection by STIV strongly induces TFS4 expression, we propose that TFS4 is part of a novel host response mechanism in archaea to combat viral infection. However, whether this response is specific to STIV or represents a more general archaeal host response to viral infection is unknown. Future experiments including mutagenesis studies of TFS4 in a crenarchaeal species that serves as host for virus infection and is amenable to genetic intervention will play an important role in elucidating the biological role of global transcription inhibition by TFS4.

## Methods

**Recombinant protein expression and purification.** *S. solfataricus* (Sso) RNAP was expressed and purified as previously described[29]. SsoTBP and SsoTFB-His were expressed and purified as previously described[51]. SsoTFS1 (sso0291) and SsoTFS4 (sso9221) genes were cloned into expression vector pET21a(+) using restriction enzymes NdeI and XhoI (NEB). Recombinant TFS1 and TFS4 were induced in Rosetta 2 (DE3) pLysS and BL21 (DE3), respectively, at a $OD_{600}$ of 0.6 by addition of 0.1 mM IPTG in enriched growth medium at 37 °C for 3 h. Cells were resuspended in 20 ml N buffer (25 mM Tris/HCl pH 8.0, 10 mM $MgCl_2$, 100 μM $ZnSO_4$, 5 mM DTT, 10% glycerol) with 50 mM NaCl (N(50), salt concentration given in parenthesis) supplemented with EDTA-free protease inhibitor tablets (Roche) and DNase I (Sigma-Aldrich). After disruption using by sonication, the cell lysate was cleared by centrifugation, incubated for 20 min at 65 °C and denatured host proteins were removed by centrifugation. The lysate was loaded onto MonoQ 5/50 GL column (GE Life Sciences). TFS1 and TFS4 were eluted by 10 ml gradient to N(1000). The fractions containing the protein were combined and concentrated using Amicon Ultra-0.5 ml (GE Life Sciences). The sample was

loaded onto Superose 12 10/300 GL column (GE Life Sciences) equilibrated in N200 buffer.

**Abortive and promoter-directed transcription assays.** In total, 20 μl samples for abortive transcription assays contained Transcription Buffer (10 mM MOPS pH 6.5, 10 mM $MgCl_2$, 250 mM NaCl, 10% glycerol, 0.05 mg BSA), 500 fmol of dsDNA template pol592/593 (homoduplex) or pol592/603 (−4 to −1 heteroduplex) (Supplementary Fig. 2a), 250 μM ApG dinucleotide, 50 μM ATP (containing [α-$^{32}$P]-ATP), 125 nM TFB, 1 μM TBP, 50 nM RNAP (and 1 μM SsoTFEα/β)[29]. Samples were incubated for 15 min at 75 °C. Reactions were stopped by transferring the sample into cold loading buffer containing formamide, and samples were heated to 95 °C. In total, 5 μl of the samples were separated on 20% polyacrylamide, 7 M Urea, 1 × TBE gels. For quantification of NTP/TFS4-concentration dependent abortive initiation, the concentration of ATP varied from 0 to 125 μM and the concentration of TFS4 varied from 0 to 250 nM. The amount of abortive initiation products ApGpA was quantified using ImageQuant TL Software (GE Healthcare). Average $K_m$ ATP for each condition was determined by curve-fitting of plotted data by nonlinear regression analysis using PRISM software (GraphPad Software, Inc.).

For promoter-directed in vitro transcription, SSV-T6 promoter template cloned into pGEM-T vector (Promega)[29] was linearised using the restriction enzyme SalI (NEB) (Supplementary Fig. 2c). Transcription reactions contained 500 μM ATP/CTP/GTP, 2.5 μM UTP (containing [α-$^{32}$P]-UTP) and 200 ng of SalI digested plasmid template, RNAP, TBP and TFB (and TFEα/β) in Transcription Buffer. Samples were incubated for 10 min at 75 °C. Reactions were stopped by transferring the sample into cold loading buffer containing formamide, and samples were heated to 95 °C. In total, 5 μl of the samples were separated on 10% polyacrylamide, 7 M Urea, 1 × TBE sequencing gels. Gels were dried for 1 h at 80 °C under vacuum, visualised using Typhoon FLA 9500 biomolecular imager (GE Healthcare) and analysed using ImageQuant TL Software (GE Healthcare).

**Electrophoretic mobility shift assay.** For pre-initiation complexes: 15 μl samples contained Transcription Buffer with 63 nM TFB, 250 nM TBP, 50 nM RNAP and 125 fmol of $^{32}$P 5′-labelled dsDNA pol592/603 template (heteroduplex) (Supplementary Fig. 2a)[15, 52]. Samples were incubated for 5 min at 65 °C before loading onto a 7% native Tris-Glycine gels (2.5% glycerol, 1 mM DTT).

For elongation complexes: $^{32}$P 5′-labelled 0.25 μM NTS-83 was annealed with 0.25 μM TS-83 and 0.75 μM RNA14. Samples containing 50 nM RNAP and scaffold template in Transcription Buffer were incubated for 5 min at 65 °C before loading onto a 5% native Tris-Glycine gels.

**RNA cleavage assays.** For transcript cleavage assays, SSV-T6 promoter fused to a 50 bp C-less cassette template was cloned into pGEM-T vector (Promega) (Supplementary Fig. 2b). The clone was linearised using the restriction enzyme SalI (NEB). Transcription reactions contained 250 μM GTP/UTP, 1.25 μM ATP (containing [α-$^{32}$P]-ATP) and 200 ng of SalI digested plasmid template, RNAP, TBP and TFB in Transcription Buffer. After 5 min at 75 °C, 250 nM of factor (TFS1, TFS4 or TFS1-AA) was added to reaction. For reactivation assays, 250 μM of CTP and ATP were additionally added. Reactions from different time points were stopped as for promoter-directed transcription assays.

**Elongation assays.** In total, 20 μM of template DNA strand (TS-83) and 60 μM of 14 nucleotide RNA primer (RNA14) were annealed in presence of 20 μM non-template DNA strand (NTS-83) (Supplementary Fig. 2d)[32]. RNAP was incubated with the annealed template in Transcription Buffer for 5 min at 65 °C. The reaction mix was incubated for further 5 min at 65 °C in the presence of 20 μg/ml heparin. The reaction mix was incubated for further 1 min at 65 °C after the addition of transcription factors (TFS1 or TFS4). Transcription reaction was started by addition of nucleotides (500 μM UTP/CTP/GTP, 2.5 μM ATP (containing [α-$^{32}$P]-ATP). Samples were incubated for 10 min at 65 °C. Reactions were stopped as for promoter-directed transcription assays.

For RNAPII elongation assays, was incubated with the annealed template in RNAPII Buffer (20 mM HEPES, pH 7.6, 100 mM KCl, 60 mM $(NH_4)_2SO_4$, 8 mM $MgSO_4$, 10 μM $ZnCl_2$, 10% Glycerol, 0.05 mg BSA) for 5 min at 20 °C. Transcription reaction was started by addition of nucleotides (500 μM UTP/CTP/GTP, 2.5 μM ATP (containing [α-$^{32}$P]-ATP). Samples were incubated for 15 min at 20 °C.

**S. acidocaldarius strains and growth.** S. acidocaldarius strain MW001 was aerobically grown in Brock media[53] with a pH of 3 at 76 °C. The media were supplemented with 0.1% (w/v) tryptone or with 0.1% (w/v) NZ-Amine and 0.2% sucrose. The growth of the cells was monitored by measurement of the optical density at 600 nm. Preparation of electrocompetent S. acidocaldarius MW001 cells and transformation of plasmid into S. acidocaldarius were carried as described by S. V Albers and colleagues[54].

**Construction of TFS1 overexpression plasmids.** saci_0171 was amplified from S. acidocaldarius genomic DNA by PCR and cloned into NcoI and BamHI sites of

pSVA1481. *saci_0171* was excised with NcoI and EagI, and was subsequently cloned into the *E. coli/Sulfolobus* shuttle vector pSVA1450, which contains *pyrEF* as marker gene[54]. The plasmid was methylated in *E. coli* EsaBC41 strain[55] and transformed into electrocompetent MW001 cells. Positive clones were selected using uracil-deficient plates[54] and confirmed by colony PCR. For overexpression, clones were grown in Brock media supplemented with 0.01% (w/v) NZ-Amine and 0.4% Sucrose/Maltose[35].

**Western blot detection of TFS4.** Polyclonal rabbit antisera against recombinant SsoTFS4 and Rpo4/7 were raised at Davids Biotechnology (Regensburg, Germany). For multiplex Immunodetection sheep-anti Alba antiserum (obtained from Malcolm White, University St. Andrews, UK) and Alexa 488 donkey anti-sheep IgG (Life Technologies) were used against Alba, and Dylight 680 conjugated donkey anti-rabbit IgG (Bethyl Laboratories, Cambridge, United Kingdom) in combination with the respective antisera was used against SsoTFS4 or Rpo4/7.

**RNAP-TEC TFS4 docking model.** The crystal structure of RNAP from *S. solfataricus* (SsoRNAP, PDB 3HKZ) and individual homology models of SsoTFS4 N-ZR and C-ZR were combined to generate a model for RNAP-TFS4. The Phyre2 model of the SsoTFS4 N-ZR and C-ZR domains was positioned on SsoRNAP independently, in an orientation consistent with the available crystal structures of Pol II bound to Rpb9 and TFIIS (PDB 3PO3) and of Pol I bound to A12.2 (PDB 4C2M). The linker between N-ZR and C-ZR was not modelled by Phyre2 and was hand-built instead, using the eukaryotic Pol I and II structures as a guide to its path within the model.

**Data availability.** The authors declare that all relevant data are available in the manuscript and Supplementary Information Files. Additional information can be obtained from the authors upon request.

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

## Acknowledgements

We would like to acknowledge all members of the RNAP laboratory, Kristine Arnvig and Michel Werner for helpful discussions and critical reading of the manuscript. We would furthermore like to acknowledge Michaela Wagner and Sonja-Verena Albers for helpful discussions of *S. acidocaldarius* over-expression constructs. Work in the UCL RNAP laboratory is funded by a Wellcome Trust Investigator Award to F.W. (079351/Z/06/Z).

## Author contributions

Conception of study: F.W., T.F. and F.B.; Experimental work: T.F., F.B., A.C.M.C. (structural modelling, RNAPII purification), R.H., M.J.Y. (infection experiments); Data analysis: T.F., F.W., A.C.M.C.; Writing the manuscript: F.W., T.F., F.B.

## Additional information

**Competing interests:** The authors declare no competing financial interests.

