## [Peer Review File · Nature Communications]

Reviewers' comments:

Reviewer #1 (Remarks to the Author):

Review of the manuscript (NCOMMS-17-08792) “The transcript cleavage factor paralogue TFS4 is a potent RNA polymerase inhibitor” by Fouqueau et al.

The manuscript by Werner and co-workers investigates the functional properties and biological role of a novel transcription factor TFS4 from *S. solfataricus*, a paralog of eukaryotic transcript cleavage factor TFIIS. Authors show that in contrast to *S. solfataricus* TFS1, an archaeal bona fide functional homolog of TFIIS that induces endonucleolytic activity of RNA polymerase (RNAP) and stimulates productive transcription, TFS4 is an effective transcriptional inhibitor that acts by destabilizing pre-initiation and initiation complexes and altering their catalytic properties. Authors demonstrated that TFS4 acts competitively with TFS1 with which it shares substantial sequence homology, and most likely the binding site in the secondary channel of RNAP. Unexpectedly, it was found that the inhibitory activity of TFS4 can also be suppressed by basal transcription initiation factor TFE which stabilizes the pre-initiation complex (PIC) and facilitates the open complex formation through interactions with RNAP clamp domain and upstream DNA. These results point to a possible allosteric link between the structural changes in the mobile elements of the secondary channel induced by TFS4 and the distant large-scale movements of the clamp domain that lead to disruption of RNAP/basal factor-DNA interactions.

Unlike TFS1 which contains an elongated loop in the C-terminal Zn-binding ribbon (C-ZR) domain with two acidic residues at the tip responsible for catalysis of RNA hydrolysis, TFS4 has a shorter loop in the C-ZR with three basic residues that are essential for the inhibitory activity. Loop swapping and residue substitution experiments revealed that the stimulatory or inhibitory activities of each factor are defined by the nature of the loop and its tip motif. This is the most striking result of this work. Importantly, under normal growth conditions, TFS4 expression in *S. solfataricus* cells is very low. However, it is strongly induced upon viral infection causing cell growth retardation. Thus, TFS4 may represent the first known example of a general transcriptional repressor in archaea that may play an important role in cellular antiviral defense strategy.

The findings are novel and provide an important contribution to the field that warrants publication in Nature Communications. The paper will be interesting for molecular biologists and microbiologists studying the organization of transcriptional machinery and its regulation in

archaea and the mechanisms of microbial host response to viral infections. However, some of the data presented are incomplete; the manuscript contains several experimental discrepancies and inconsistencies that should be resolved. Some of the claims are not directly supported by experimental evidence. Additional experiments may be required to strengthen the paper and make it acceptable for publication. Below are my critiques, comments, and suggestions.

Major critique

1. The first problem of this work is the lack of direct evidence that would support authors' assertion that TFS4 inhibition has two components: a destabilization of RNAP-nucleic acid complexes at high factor/RNAP ratios, and a catalytic inhibition based on interference with the active site of RNAPs that can be observed at low factor concentrations. The first claim is confirmed by direct data based on EMSAs and to some extent by the results of abortive initiation assays using the minimal archaeal PIC complex prepared on SSV1 T6 promoter DNA (Fig. 3, panels a, b, e, f). Additional DNase or permanganate footprinting data would help strengthen the conclusion. However, the second claim is supported only by an indirect data. First, authors show that TFS4 inhibits productive transcription in promoter-directed runoff transcription assay at factor concentrations substoichiometric to RNAP (Fig. 4a). It appears that >50% inhibition is observed at the TFS4/RNAP molar ratio of 1:10. At the same time, the decrease of the runoff formation is not accompanied by accumulation of any RNAs shorter than 71 nt. Similar inhibitory effect was observed during the extension of RNA primer using a DNA/RNA scaffold template in transcription elongation assay (Fig. 4c). Unfortunately, in both assays, the autoradiograms only show transcripts >40 nt and >60 nt, respectively. Thus, it appears that TFS4 inhibits transcription in "all or nothing" fashion. Since TFS4 also inhibits the multiround abortive synthesis (although at 2-fold excess over RNAP), and even destabilizes the elongation complex (Fig 4e), it is possible that the factor acts during the PIC or the open complex formation, or at a subsequent step prior to substrate polymerization. To prove that TFS4 indeed inhibits the catalytic step, authors should analyze whether V_{max} or apparent K_m values are affected, using a stepwise RNA extension assay with different concentrations of NTPs. Alternatively, could TFS4 act as a termination factor? Experiments with immobilized TEC using His-tagged RNAP would help address this question.

2. My second concern is the demonstration of the biological role of TFS4. Fig. 6 clearly shows that STIV virus infection dramatically increases the level of TFS4 expression and slows down/inhibits cell growth. However, the direct role of TFS4 overexpression in this process has not been demonstrated. Instead, authors show the inhibitory effect of a hybrid factor TFS1-tip4 in the growth of a related organism *S. acidocaldarius*. While these observations are consistent with the proposed role TFS4, it would be nice to have more direct data to support this view. Will it be possible to construct an FTS4-deletion strain of *S. solfataricus* and analyze its growth curves with and without SITV infection? Alternatively, could TFS4 be overexpressed in *S. solfataricus* using a compatible plasmid vector under inducing and non-inducing conditions (like

those shown in Fig. 6d and 6e)?

Minor comments.

1. Introduction, p. 4, 2nd para, 2nd sentence. Figures 1 and S1 do not show any information on RPB9 activity. Fig. 1 shows the evolutionary conservation of TFIIS-homolog and -paralog proteins, whereas Fig. S1 shows the sequence alignment of TFS4 and phylogenetic distribution of TFS paralogues in Sulfolobales.

2. Introduction, p. 4, 2nd para, 3rd sentence. Reference 23 is incorrect: Langer and Zillig 1993 did not report on the function of A12.2 in RNAPI and C11 in RNAPIII.

3. Introduction, p. 4, 2nd para, 6th and 7th sentence. Add a reference. Authors should also cite Laptenko et al., EMBO J, 2003, where the structural model for the bacterial Gre-RNAP complex has been presented, and the role of the GreA's tip acidic residues in catalysis has been demonstrated.

4. Introduction, p.5, 1st para, 2nd sentence. A typo: ...three lysine residues.

5. Introduction, p.5, 1st para, last sentence. The statement is unclear; please rephrase.

6. Results, p.6, 1st para, 3rd sentence. Fig. 1 does not show the sequence or the functional properties of TFS2. The sequence alignment of *S. solfataricus* TFS2 and TFS3 homologs should be added to Fig. 1 or Fig. S1.

7. Results, p.6, 1st para, 6th and 7th sentence. Do authors mean to say that TFS2 and TFS3 are not RNAP-binding factors? Maybe the choice of TFS4 (over TFS2 and TFS3) for this study should be explained in more details. Also, the loop (tip) of TFS4 is 8 residues shorter than that of TFS1, so it may not reach the catalytic site of RNAP.

8. Results, p.7, 2nd para, 4th sentence, and Fig. 2b. Why were the 50 nt and 49 nt cleavage products not extended? The reaction mix contained the same three substrates: ATP, GTP, and UTP? I would expect extension reaction to be more efficient than RNA cleavage.

9. Results, p.8, 1st para, 4th sentence and Fig. 3a. The efficiency of PIC formation is rather low. Was heparin added to the reaction prior to electrophoresis by EMSA?

10. Results, p.8, 1st para, 7th sentence, and Fig. 3b. What was the concentration of TFS4 in the assay shown in Fig 3b?

11. Results, p.9, 2nd sentence from the bottom and Fig. 4a. TFS4 inhibits runoff synthesis by ~50% even when added at substoichiometric concentrations to RNAP (0.1:1). Note that RNAP was present in ~2-6 fold excess over DNA. Could this inhibitory effect be because TFS4 is specific (has a high binding affinity) to PIC and has a low affinity to free RNAP? Also, the lower part of the gel autoradiogram is not visible. Could the synthesis of smaller (5-40 nt) RNA products be affected by TFS4?

12. Results, p.10, 1st para, last sentence and Fig. 4d and 4e. The efficiency of TEC formation is very low: only a minor fraction of the DNA/RNA scaffold forms a stable complex with RNAP (a slow-migrating band of TEC on the gel in panels d and e). It is difficult to assess and compare the effects of TFS1 and TFS4 on the stability of TEC. The evidence for inhibition of elongation complex formation by TFS4 is not very convincing. What is the reason for such a low yield of TEC?

13. Discussion, p.15, 2nd and 3rd sentence. Similar to the effect of mutations of acidic tip in TFIIS and TFS1, mutation of the tip residues in *E. coli* GreA (including the catalytic Asp and Glu) also convert Gre factors into strong inhibitors of transcription elongation that lead to a dominant lethal phenotype (see Laptenko et al., EMBO J, 2003).

14. Discussion, p.15, 4th sentence from the bottom. The inhibition of RNAP catalysis by TFS4 was not shown directly (see Major critique 1), it's only a conjecture.

15. Discussion, p.16, 1st para, last sentence. The expression of TFS4 during the stationary phase of uninfected *S. solfataricus* was not shown.

16. Methods, p. 18 and 19, Abortive and promoter-directed transcription assays, and elongation assays. Reaction conditions are not described clearly. What is the size of promoter DNA fragment used in the assays? The final concentrations of DNA and RNAP in all reactions should be indicated.

17. Fig. 3. Panels g and h. The indication for the presence/absence of factors in the reaction appears to be incorrect: lane 2 is supposed to be a positive control with both TFB and TBP present (compare lanes 2 and 3 in panels e and f).

18. Fig. 7c. The depiction of the secondary channel (funnel) in the schematic representation of Sso RNAP is incorrect. The funnel should be placed on the downstream side of the protein facing the two Mg ion, not the 5'-end of the nascent RNA.

19. Supplementary Fig. S4. Incorrect legend. TFE α/β should be indicated as present (+) only at the right side of panels a and b.

Reviewer #2 (Remarks to the Author):

Review of Fouqueau et al. for Nature Communications

Fouqueau et al. describe the discovery and characterization of an interesting new transcription factor present in the archaea *Sulfolobus sulfataricus* that is evolutionarily but not functionally related to the well-known transcript cleavage/fidelity factor TFIIS (TFS1 in archaea). This new factor, called TFS4, was discovered bioinformatically along with TFS2 and TFS3, which remain uncharacterized. TFS4 appears to have an N-terminal RNAP-binding domain like TFS1, but the C-terminal domain that interacts in the secondary channel has a different structure and contains 4 Lys residues near its tip whereas two acidic residues are located near the tip in TFS1 (and in TFIIS and in the structurally unrelated bacterial Gre factors). These acidic residues are responsible for chelating a second Mg ion in the RNAP active site that is required for transcript cleavage. The authors establish that TFS4 is a transcription inhibitor and show that, although not expressed in typical growth conditions, it is induced by infection with a *Sulfolobus* virus. The authors convincingly establish that TFS4 inhibits nucleotide addition in RNAP and also appears to destabilize both elongation complexes and preinitiation complexes formed with TBP, TFB, and TFE (although TFE may be indirectly competitive with TFS4). Although a complete picture of the *in vivo* function of TFS4 does not emerge from the findings, this is a remarkably well-rounded characterization of a transcription factor for the manuscript that first describes its discovery. The work will be of general interest in the transcription field and to researchers in general interested in gene regulation (this type of inhibitor is not restricted to archaea as bacterial examples related to Gre factors have been found, and it would be unsurprising if eukaryotic examples remain to be discovered). I believe the manuscript is largely suitable for publication and have just a few larger questions and a list of comments for the authors to consider in a final revision.

Major points

1. The authors suggest that the inhibitory effect of TFS4 results from an allosteric effect of TFS4 on RNAP or ECs, that to say binding of TFS4 in the secondary channel alters the conformation of RNAP in a way that inhibits PIC formation and transcript elongation and promotes EC dissociation. This is certainly one model to be considered and I understand why the authors may favor it based on effects on PIC formation and EC dissociation. However, binding of proteins in

the secondary channel can also be expected to inhibit NTP binding directly by blocking the path of NTP entry to the active site and somewhat less obviously can potentially inhibit DNA binding on the downstream side of the active site. Thus, an alternative model for TFS4 action could be simple direct competition for substrate NTP or DNA, yet the authors do not mention this possibility let alone exclude it experimentally. At a minimum, the alternative model should be described. It would be desirable to also include simple experiments to test the direct competition models. Do low concentrations of TFS4 raise the apparent K_{ntp} for substrates during elongation? Do changes in NTP and TFS4 concentration give competitive effects? Would higher concentrations of DNA lessen the inhibitory effect of TFS4 on EC dissociation?

2. The effects of virus infection on TFS4 expression are interesting and consistent with the authors descriptions, but the results stop short of directly defining the role of TFS4 in phage infection. Is there some reason TFS4 expression can't be blocked in *Sulfolobus*? (eg, a knockout mutant or some sort of antisense RNA interference expt?). I don't know the system well enough to know how hard this would be, but it would be very interesting to know if the virus infection is altered in a TFS4 knockout. If that's a doable experiment, then I'd recommend trying to include it. If it's not a doable experiment, then I'd recommend setting up the last section of the results by explaining that it would be the ideal experiment but is not feasible in *Sulfolobus* and therefore an alternative approach was used (to help readers understand why the approach used was selected).

3. Can the authors say anything about the relative strengths of TFS1 and TFS4 binding. Do they compete with roughly the same affinity or does one bind much tighter than the other. It would be particularly interesting if TFS4 bound much tighter than TFS1 because TFS1 must bind weakly enough to allow NTP binding and tighter binding of TFS4 would make it an effective block to NTP entry. The authors perform competition experiments so this seems like an entirely reasonable question to address in this manuscript. The data may already exist or a simple extension of experiments already performed would address this quite interesting question.

Minor points

1. Abstract, third line and several other places in the manuscript. "While" is best reserved to mean "contemporaneously" - "Although" is a better word choice here.

2. Abstract line 8. Although I agree an "allosteric" model is a reasonable explanation, since the experiments do not provide unambiguous discrimination against a direct competition model it might be safest to be more conservative in the abstract.

3. p3 li5. The "secondary channel" is generally accepted nomenclature for bacterial RNAP (and perhaps for archaeal RNAP - I'm uncertain in that case), but for eukaryotic RNAPII the terms

“pore” and “funnel” are the generally accepted nomenclature for the outer and inner parts of the secondary channel. It might help target a broader audience to explain both sets of terms here.

4. p3 li9 should be a comma after phase

5. p3 li11 It is more accurate from a cellular perspective to say that DNA moves backwards through RNAP than that RNAP moves backwards on DNA since the DNA is generally more mobile than the RNAP. Either frame of reference is correct, however.

6. p3 third line from bottom “urgent” implies a need to act quickly. Perhaps “strong” would be a better choice.

7. p4 li17 Generally, cleavage factors are thought to stabilize binding of the second Mg rather than alter the position of binding and it might be better to substitute stabilize for allow here.

8. p4 li22 The first sentence of this para is a little awkward. Perhaps something like “Here we describe the discovery of a fascinating multiplication of tfs-like genes in archaea and present the characterization of two of them, TFS1 and TFS4, that display opposite stimulatory and inhibitory effects on transcription.

9. p4 third line from the bottom. This itemized list is not parallel; (i) destabilizes, (ii) inhibits, (iii) no verb, (iv) no verb. Rephrase to create a parallel construction.

10. p5 li3 typo “tree” should be “three”

11. p6 li16 Is it possible that His residues substitute for the missing Cys residues in the ZR domains?

12. p7 li7 Fig. 2a does not show the template referred to in this citation. It would be very helpful to readers to depict the template structure/sequence in key segments and a brief experimental schematic.

13. p8 li15 It is unclear to me why the inhibition of PIC formation necessarily means an allosteric effect. Couldn't parts of TFS4 be located where they directly compete with DNA?

14. p9 li11 The difference in salt concentrations for high vs low is not particularly large. Is Cl even the relevant counterion for in vivo. For many organisms Cl is excluded and can compete for protein-phosphate ionic interactions.

15. p10 li4 Wasn't an effect of TFS4 on inhibition of elongation shown in Fig. 2?

16. p11 last line. This is a point where it's particularly unclear why the simple steric inhibition of NTP binding isn't considered as an explanation for inhibitory effects on elongation.

17. p12 li14 use of two "which" in the independent clause is confusing.

18. p15 li7 There also are data on a Gre mutant in which substitution of one or two of the acidic residues turns it into an inhibitor. I apologize for not looking up the citation, but I think it's work from Arkady Mustaev, and would be worth citing here.

19. p15 li11 Again here, it's unclear why the direct competition for NTP binding isn't considered.

20. Fig. 4 Why is the formation of ECs so inefficient? What percentages of DNA and RNAP are formed into EC? Since it's hard to see appearance of DNA released from EC upon TFS4 binding, the conclusion that TFS4 disrupts EC is not particularly well supported by the experiment. Isn't it possible that DNA remains bound and TFS4 binding just causes the EC to smear out during electrophoresis?

21. Fig. 5d and 5e Is the x-axis correctly labeled? It seems like it should be TFS4/RNAP ratio. Also, maybe don't use gray for the triple A mutant – it's hard to distinguish from black on the figure.

**Detailed response to reviewer's comments**

Reviewer #1 (Remarks to the Author):

Review of the manuscript (NCOMMS-17-08792) "The transcript cleavage factor paralogue TFS4 is a
potent RNA polymerase inhibitor" by Fouqueau et al.

The manuscript by Werner and co-workers investigates the functional properties and biological role of
a novel transcription factor TFS4 from *S. solfataricus*, a paralog of eukaryotic transcript cleavage factor
TFIIS. Authors show that in contrast to *S. solfataricus* TFS1, an archaeal bona fide functional homolog
of TFIIS that induces endonucleolytic activity of RNA polymerase (RNAP) and stimulates productive
transcription, TFS4 is an effective transcriptional inhibitor that acts by destabilizing pre-initiation and
initiation complexes and altering their catalytic properties. Authors demonstrated that TFS4 acts
competitively with TFS1 with which it shares substantial sequence homology, and most likely the
binding site in the secondary channel of RNAP. Unexpectedly, it was found that the inhibitory activity of
TFS4 can also be suppressed by basal transcription initiation factor TFE which stabilizes the
preinitiation
complex (PIC) and facilitates the open complex formation through
interactions with RNAP clamp domain and upstream DNA. These results point to a possible allosteric
link between the structural changes in the mobile elements of the secondary channel induced by TFS4
and the distant large-scale movements of the clamp domain that lead to disruption of RNAP/basal
factor-DNA interactions.

Unlike TFS1 which contains an elongated loop in the C-terminal Zn-binding ribbon (C-ZR) domain with
two acidic residues at the tip responsible for catalysis of RNA hydrolysis, TFS4 has a shorter loop in
the
C-ZR with three basic residues that are essential for the inhibitory activity. Loop swapping and residue
substitution experiments revealed that the stimulatory or inhibitory activities of each factor are defined
by the nature of the loop and its tip motif. This is the most striking result of this work. Importantly,
under normal growth conditions, TFS4 expression in *S. solfataricus* cells is very low. However, it is
strongly induced upon viral infection causing cell growth retardation. Thus, TFS4 may represent the
first known example of a general transcriptional repressor in archaea that may play an important role in
cellular antiviral defense strategy.

**The findings are novel and provide an important contribution to the field that warrants publication in**

**Nature Communications. The paper will be interesting for molecular biologists and microbiologists**
**studying the organization of transcriptional machinery and its regulation in archaea and the**
**mechanisms of microbial host response to viral infections.** However, some of the data presented are
incomplete; the manuscript contains several experimental discrepancies and inconsistencies that
should be resolved. Some of the claims are not directly supported by experimental evidence.
Additional experiments may be required to strengthen the paper and make it acceptable for
publication. Below are my critiques, comments, and suggestions.

Major critique

1. The first problem of this work is the lack of direct evidence that would support authors' assertion
that TFS4 inhibition has two components: a destabilization of RNAP-nucleic acid complexes at high
factor/RNAP ratios, and a catalytic inhibition based on interference with the active site of RNAPs that
can be observed at low factor concentrations. The first claim is confirmed by direct data based on
EMSAs and to some extent by the results of abortive initiation assays using the minimal archaeal PIC
complex prepared on SSV1 T6 promoter DNA (Fig. 3, panels a, b, e, f). Additional DNase or
permanganate footprinting data would help strengthen the conclusion.

*We have followed the recommendation of reviewer-1 and carried out permanganate footprinting*
*experiments. The results are shown in supplementary figure 2. We use the same SSV1 T6 DNA*
*template as in the EMSA experiments shown in figure 3, where the noncomplementary T-residue in the*
*4-bp bubble (highlighted in red) at register -1 serves as positive control. In the presence of both TBP*
*and TFB additional permanganate reactivity can be detected at the -5, -7 and -12 positions (highlighted*
*with asterisks on the DNA sequence), which is further enhanced by the addition of TFE alpha/beta. The*
*inclusion of TFS4 abolishes the permanganate reactivity, in good agreement with the PIC destabilizing*
*activity of TFS4. The addition of TFE alpha/beta counteracts this TFS4 inhibition, also in perfect*
*agreement with the results of our EMSA assays shown in figure 3.*

However, the second claim is supported only by an indirect data. First, authors show that TFS4 inhibits
productive transcription in promoter-directed runoff transcription assay at factor concentrations
substoichiometric to RNAP (Fig.4a). It appears that >50% inhibition is observed at the TFS4/RNAP
molar ratio of 1:10. At the same time, the decrease of the runoff formation is not accompanied by
accumulation of any RNAs shorter than 71 nt. Similar inhibitory effect was observed during the
extension of RNA primer using a DNA/RNA scaffold template in transcription elongation assay (Fig.
4c). Unfortunately, in both assays, the autoradiograms only show transcripts >40 nt and >60 nt,

respectively. Thus, it appears that TFS4 inhibits transcription in “all or nothing” fashion. Since TFS4
 also inhibits the multiround abortive synthesis (although at 2-fold excess over RNAP), and even
 destabilizes the elongation complex (Fig 4e), it is possible that the factor acts during the PIC or the
 open complex formation, or at a subsequent step prior to substrate polymerization.

We completely agree with reviewer-1, TFS4 does not impair transcription processivity, which would
 result in the accumulation of short, partial products that we never observed. In order to provide further
 evidence for this we have included a ‘longer’ gel below this paragraph. These figures demonstrate that
 no smaller (nor longer) transcripts are synthesized, and the minimal size of transcripts resolved on the
 new gels is about 9 nt. We do not feel that it is necessary to include the ‘extended’ gel in the figure of
 the article due to space constraints – but are happy to reconsider if the editor deems it critical. We
 concede to reviewer-1’s point of view that - in theory - a TFS4-based interference with transcription
 initiation or -elongation complex stability and integrity would impair transcription activity. However, the
 catalytic inhibition occurs at much *lower* TFS4 concentrations than are necessary to destabilise these
 complexes, as shown in our direct binding EMSA experiments. The molecular basis for the catalytic
 inhibition could be interference with substrate NTP binding and/or phosphodiester bond formation.

To prove that TFS4 indeed inhibits the catalytic step, authors should analyze whether V_{max} or
apparent K_m values are affected, using a stepwise RNA extension assay with different concentrations
of NTPs.

In order to elaborate on the underlying molecular mechanism of catalytic inhibition, we followed
reviewer-1's advice and carried out single NTP addition experiments using abortive transcription
assays over a range of NTP substrate concentrations, and using a range of TFS4 concentrations. We
show the results as double reciprocal plot in a *Lineweaver-Burke* fashion in figure 3. The results clearly
show that the trend lines corresponding to different TFS4 concentrations intersect the $(1/v)$ Y-axis at
the same point, while the intersections with the x-axis $(1/[S])$ differ at varying TFS4 concentrations. This
is strongly *indicative of a competitive inhibition*, and in molecular terms likely reflects that TFS4 binding
interferes with NTP substrate binding, or interferes with the access of NTP substrates to their binding
site(s). This is in perfect agreement with the proposed binding site of TFS4 in the NTP entry pore, and
with the binding modes of eukaryotic (TFIIS) and bacterial (Gre) cleavage factors in the secondary,
NTP entry, channels of their cognate RNA polymerases. The results have been incorporated into figure
3h and are discussed on page 9:

*'To further elaborate on the underlying mechanism of inhibition we tested whether the TFS4 inhibition*
*varied with NTP concentrations. We carried out abortive initiation assays at a range of NTP substrate-*
*and TFS4 concentrations, and the results are illustrated as double reciprocal plot in Figure 3g (primary*
*data in supplementary Fig. 6a and b). The results show that TFS4 acts as competitive inhibitor*
*interfering with NTP binding. At 250 nM TFS4 (5-fold excess over RNAP) the apparent K_m for NTP is*
*increased ~30-fold from $27 \pm 3 \mu M$ to $870 \pm 81 \mu M$.*

Alternatively, could TFS4 act as a termination factor? Experiments with immobilized TEC using His-
tagged RNAP would help address this question.

Transcription termination factors classically disrupt *elongation complexes* in a gene sequence-
dependent context, either at the 3' of the gene to facilitate RNA3' formation or in the 5' region of a gene
as mechanism of regulation (termination-antitermination). Since TFS4 also targets transcription
*initiation complexes*, and since we have no evidence nor theoretical mechanism to suggest any
sequence-dependent activity we do not favor to use the term *termination factor* for TFS4.

2. My second concern is the demonstration of the biological role of TFS4. Fig. 6 clearly shows that
STIV virus infection dramatically increases the level of TFS4 expression and slows down/inhibits cell
growth. However, the direct role of TFS4 overexpression in this process has not been demonstrated.
Instead, authors show the inhibitory effect of a hybrid factor TFS1-tip4 in the growth of a related
organism *S. acidocaldarius*.

The choice of *S. acidocaldarius* is not only a technical one – it is amenable to genetics and vector-
based overexpression is possible – but chiefly because this organism does not encode any TFS4
homologues and therefore provides the perfect ‘clean’ strain background for an ‘ectopic’
overexpression experiment.

While these observations are consistent with the proposed role TFS4, it would be nice to have more
direct data to support this view. Will it be possible to construct an TFS4-deletion strain of *S. solfataricus*
and analyze its growth curves with and without SITV infection?

*S. solfataricus* P2 and its derivative strain 2-2-12 do not allow gene knockouts.

Alternatively, could TFS4 be overexpressed in *S. solfataricus* using a compatible plasmid vector under
inducing and noninducing conditions (like those shown in Fig. 6d and 6e)?

There are no reliable vector tools for the controlled overexpression of recombinant proteins in *S.*
*solfataricus* P2 and 2-2-12.

Minor comments.

1. Introduction, p. 4, 2nd para, 2nd sentence. Figures 1 and S1 do not show any information on RPB9
activity. Fig. 1 shows the evolutionary conservation of TFIS-homolog and -paralog proteins, whereas
Fig. S1 shows the sequence alignment of TFS4 and phylogenetic distribution of TFS paralogues in
Sulfolobales.

We introduced the suggested change.

2. Introduction, p. 4, 2nd para, 3rd sentence. Reference 23 is incorrect: Langer and Zillig 1993 did not
report on the function of A12.2 in RNAPI and C11 in RNAPIII.

We introduced the suggested change.

3. Introduction, p. 4, 2nd para, 6th and 7th sentence. Add a reference. Authors should also cite Laptenko et al., EMBO J, 2003, where the structural model for the bacterial Gre-RNAP complex has been presented, and the role of the GreA's tip acidic residues in catalysis has been demonstrated.

We introduced the suggested reference.

4. Introduction, p.5, 1st para, 2nd sentence. A typo: ...three lysine residues.

We introduced the suggested change.

5. Introduction, p.5, 1st para, last sentence. The statement is unclear; please rephrase.

We rephrased the sentence to: 'Interestingly, infection of *Sso* with the *Sulfolobus* Turreted Icosahedral Virus (STIV) induces TFS4 expression suggesting that TFS4 plays a role in the antiviral host response and defence.'

6. Results, p.6, 1st para, 3rd sentence. Fig. 1 does not show the sequence or the functional properties of TFS2. The sequence alignment of *S. solfataricus* TFS2 and TFS3 homologs should be added to Fig. 1 or Fig. S1.

We introduced the suggested change into Figure S1

7. Results, p.6, 1st para, 6th and 7th sentence. Do authors mean to say that TFS2 and TFS3 are not RNAP-binding factors? Maybe the choice of TFS4 (over TFS2 and TFS3) for this study should be explained in more details. Also, the loop (tip) of TFS4 is 8 residues shorter than that of TFS1, so it may not reach the catalytic site of RNAP.

This study if focused on the TFS variant that is most closely related to the canonical archaeal transcript cleavage factors *yet* has functionally diverged in intriguing ways. Studies with TFS2 and -3 are underway and will be published elsewhere in due time. The fact that the computational homology model suggests that the tip domain is shorter in TFS4 compared to TFS1 is interesting, but no stringent

conclusions concerning the depth of penetration into the RNAP active site can be drawn based on
sequence conservation and homology modelling alone. Thus it is not inconceivable that the TFS4 tip is
drawn deeper into the NTP entry channel (via electrostatic interactions with the three lysines) than the
TFS1 tip, which would support the notion that the insertion of TFS4 leads to conformational changes in
RNAP (like Gfh1), while the insertion of TFS1 less so, or not at all (like Gre).

8. Results, p.7, 2nd para, 4th sentence, and Fig. 2b. Why were the 50 nt and 49 nt cleavage products
not extended? The reaction mix contained the same three substrates: ATP, GTP, and UTP? I would
expect extension reaction to be more efficient than RNA cleavage.

The generated transcript size reflects the equilibrium between synthesis and cleavage. Under the given
reaction conditions ($[GTP]=[UTP]=250\ \mu\text{M}$ and $[ATP]=1.25\ \mu\text{M}$) and TFS concentrations the cleavage
products prevail in the presence of cleavage factor TFS1. The schematic in Figure 2A has been
corrected with respect to the ATP concentration. The K_m of NTPs in archaeal RNAPs is approximately
30 micro molar, which at least in part explains why cleavage 'wins' over extension. But more
importantly, the negative control in this experiment is the TFS1 double alanine substitution mutant
(TFS1-AA) which is not able to induce transcript cleavage, and likewise the addition of TFS4 does not
induce cleavage – leaving the extended transcript the dominant transcript species. Figure 2 panel D
shows the reactivation of stalled complexes under conditions where all four NTPs are present. In the
absence of cleavage factors the Sso RNAP cannot resume elongation (see below), while the addition
of TFS1 (but not TFS1-AA or TFS4) efficiently reactivates the stalled complexes, which highlights the
importance of transcript cleavage factors in archaea. Just to be clear on this point, in the absence of
CTP the fully Watson-Crick base pairing-compliant RNA can only be extended to the A residue at
register +50. The major extension product in our assay is 51 nt long and arises from mis-incorporation
of a single non-complementary nucleotide at the 3' terminus. This is a well-known phenomenon that
occurs in archaeal RNAP (Fouqueau et al., NAR 2013) and RNA polymerase II (Jeon et al., JBC 1996),
and requires transcript cleavage factors for reactivation. We have introduced a sentence to clarify the
details into the revised manuscript:

*'Initiation of transcription by the addition of RNAP and the two essential basal transcription factors TBP*
*and TFB in the presence of ATP, GTP and UTP resulted in TECs stalling after synthesizing a 51 nt*
*transcript. This transcript is generated by the misincorporation of a single nucleotide at the 3' end of the*
*50 nt C-less transcript, a phenomenon observed in archaeal RNAP⁸ as well as eukaryotic RNAPII²⁹.*

... and at the bottom of the paragraph:

*'Without cleavage factor only a negligible fraction of the 51 nt transcript (containing the misincorporated*
*3'-end nucleotide) could be extended, in agreement with the cleavage assays results discussed above.'*

9. Results, p.8, 1st para, 4th sentence and Fig. 3a. The efficiency of PIC formation is rather low. Was
heparin added to the reaction prior to electrophoresis by EMSA?

We beg to differ with reviewer-1, the efficiency of PIC formation in our experiments is not low but
typical for archaeal PIC (Werner and Weinzierl, 2002 and 2005, Blombach et al., 2015, Sheppard et al.,
2016). The addition of heparin is very important in these experiments to suppress the TBP/TFB factor-
independent binding of RNAP to the promoter template.

10. Results, p.8, 1st para, 7th sentence, and Fig. 3b. What was the concentration of TFS4 in the assay
shown in Fig 3b?

The concentration of TFS4 is 100 nM. This information was added to the legend of Figure 3.

11. Results, p.9, 2nd sentence from the bottom and Fig. 4a. TFS4 inhibits runoff synthesis by ~50%
even when added at substoichiometric concentrations to RNAP (0.1:1). Note that RNAP was present in
~2-6 fold excess over DNA. Could this inhibitory effect be because TFS4 is specific (has a high binding
affinity) to PIC and has a low affinity to free RNAP?

Despite several attempts, it has not been possible to measure the direct binding affinity of TFS1 or
TFS4 to RNAP in the binary complexes, Or to the PIC or to the TEC, and we have therefore been very
careful not to make any statement about absolute binding affinities of TFS1 and -4 throughout the
manuscript. We can only pass a semi-quantitative judgment on relative binding affinities based on
competition between TFS1 and -4 during PIC formation (figure 3b), which suggest that TFS4 binds with
higher affinity to the PIC than TFS1. Similarly, we have measured the TFS1 concentration-dependent
relief of TFS4 inhibition in transcription elongation assays (Supplementary figure 8), and found that the
apparent IC₅₀ for TFS1 is 230 nM at a TFS4 concentration of 100 nM. This suggests that the affinity of
TFS4 to RNAP is higher than the affinity of TFS1 for RNAP, in good agreement with the EMSA PIC
competition experiments.

Also, the lower part of the gel autoradiogram is not visible. Could the synthesis of smaller (5-40 nt)
RNA products be affected by TFS4?

As stated above, TFS4 does *not* impair transcription processivity and does not result in the
accumulation of partial RNA products. In order to provide further evidence for this we have included a
'longer' gel in this document (see above), but not in the revised manuscript file due to space
constraints.

12. Results, p.10, 1st para, last sentence and Fig. 4d and 4e. The efficiency of TEC formation is very
low: only a minor fraction of the DNA/RNA scaffold forms a stable complex with RNAP (a slow
migrating band of TEC on the gel in panels d and e). It is difficult to assess and compare the effects of
TFS1 and TFS4 on the stability of TEC. The evidence for inhibition of elongation complex formation by
TFS4 is not very convincing. What is the reason for such a low yield of TEC?

We have followed reviewer-1s advice and repeated the TEC EMSA assays. The new optimized binding
experiments show now very clearly (i) a superhift of the RNAP-DNA-RNA upon the addition of TFS1
reflecting RNAP-DNA-RNA-TFS1 complex formation, and (ii) that the addition of TFS4 disrupts the
RNAP-DNA-RNA complex. In order to allow a direct comparison we have used the same factor:RNAP
ratios for TFS1 and -4.

13. Discussion, p.15, 2nd and 3rd sentence. Similar to the effect of mutations of acidic tip in TFIIIS and
TFS1, mutation of the tip residues in E. coli GreA (including the catalytic Asp and Glu) also convert Gre
factors into strong inhibitors of transcription elongation that lead to a dominant lethal phenotype (see
Laptenko et al., EMBO J, 2003).

We have altered the following sentence in the discussion to include reviewer-1's suggestion: 'The
catalytically inactive cleavage factor mutant TFS1-AA impairs processivity *in vivo* and *in vitro* in a
fashion that is distinct from TFS4 (Fig. 4a), in direct comparison with the cognate human TFIIIS-AA and
bacterial GreA-AA mutant variants that also have inhibitory activities and a dominant negative lethal
phenotype^{39,40}.

14. Discussion, p.15, 4th sentence from the bottom. The inhibition of RNAP catalysis by TFS4 was not
shown directly (see Major critique 1), it's only a conjecture.

Our new experiments led additional credence to the hypothesis of catalytic inhibition. We modified the
text based on our observations that TFS4 lowers the Km for NTP substrates to the following:

*'The TFS4 inhibition appears to have two components, a catalytic inhibition based on interference with*
*the access of NTP substrate to the active site of RNAPs.*

15. Discussion, p.16, 1st para, last sentence. The expression of TFS4 during the stationary phase of
uninfected *S. solfataricus* was not shown.

There is no evidence to suggest the expression of TFS4 mRNA or TFS4 protein under conditions *other*
*than* during STIV virus infection. We added a panel comparing the TFS4 immunoreactivity in biomass
from uninfected *S. solfataricus* P2 cultures isolated in the exponential and stationary growth phase,
respectively, to **Supplementary Figure 12a**. These results clearly show that TFS4 expression is not
detectable in either growth phase. We also rephrased the sentence to the following: "Starvation and
oxidative stress induce dramatic depletion of TFE β . It is therefore possible that the inhibitory effect of
TFS4 on transcription initiation is more dominant under certain growth conditions".

16. Methods, p. 18 and 19, Abortive and promoter-directed transcription assays, and elongation
assays. Reaction conditions are not described clearly. What is the size of promoter DNA fragment used
in the assays? The final concentrations of DNA and RNAP in all reactions should be indicated.

We have amended these omissions. The transcription templates are described in the published
literature (Hirtreiter et al., NAR 2010 and Blombach et al., eLife 2015). DNA template sequences used
in this study are now listed in Supplementary Fig.12. The concentrations of RNAP and DNA is explicitly
stated in the methods section in the revised manuscript.

17. Fig. 3. Panels g and h. The indication for the presence/absence of factors in the reaction appears
to be incorrect: lane 2 is supposed to be a positive control with both TFB and TBP present (compare
lanes 2 and 3 in panels e and f).

We corrected the mistake.

18. Fig. 7c. The depiction of the secondary channel (funnel) in the schematic representation of Sso
RNAP is incorrect. The funnel should be placed on the downstream side of the protein facing the two
320 Mg ion, not the 5'-end of the nascent RNA.

We have altered the schematic drawing accordingly.

19. Supplementary Fig. S4. Incorrect legend. TFE/ should be indicated as present (+) only at the right
side of panels a and b.

We introduced the suggested change.

**Reviewer #2 (Remarks to the Author):**

Review of Fouqueau et al. for Nature Communications

Fouqueau et al. describe the discovery and characterization of an interesting new transcription factor
present in the archaea *Sulfolobus sulfataricus* that is evolutionarily but not functionally related to the
well-known transcript cleavage/fidelity factor TFIIS (TFS1 in archaea). This new factor, called TFS4,
was discovered bioinformatically along with TFS2 and TFS3, which remain uncharacterized. TFS4
appears to have an N-terminal RNAP-binding domain like TFS1, but the C-terminal domain that
interacts in the secondary channel has a different structure and contains 4 Lys residues near its tip
whereas two acidic residues are located near the tip in TFS1 (and in TFIIS and in the structurally
unrelated bacterial Gre factors). These acidic residues are responsible for chelating a second Mg ion in
the RNAP active site that is required for transcript cleavage. The authors establish that TFS4 is a
transcription inhibitor and show that, although not expressed in typical growth conditions, it is induced
by infection with a *Sulfolobus* virus. The authors convincingly establish that TFS4 inhibits nucleotide
addition in RNAP and also appears to destabilize both elongation complexes and preinitiation
complexes formed with TBP, TFB, and TFE (although TFE may be indirectly competitive with TFS4).
Although a complete picture of the in vivo function of TFS4 does not emerge from the findings, **this is a**
**remarkably well-rounded characterization of a transcription factor for the manuscript that first describes**
**its discovery. The work will be of general interest in the transcription field and to researchers in general**
**interested in gene regulation (this type of inhibitor is not restricted to archaea as bacterial examples**
**related to Gre factors have been found, and it would be unsurprising if eukaryotic examples remain to**
**be discovered). I believe the manuscript is largely suitable for publication and have just a few larger**
**questions and a list of comments for the authors to consider in a final revision.**

Major points

1. The authors suggest that the inhibitory effect of TFS4 results from an allosteric effect of TFS4 on
RNAP or ECs, that to say binding of TFS4 in the secondary channel alters the conformation of RNAP
in a way that inhibits PIC formation and transcript elongation and promotes EC dissociation. This is
certainly one model to be considered and I understand why the authors may favor it based on effects
on PIC formation and EC dissociation. However, binding of proteins in the secondary channel can also

be expected to inhibit NTP binding directly by blocking the path of NTP entry to the active site and
somewhat less obviously can potentially inhibit DNA binding on the downstream side of the active site.
Thus, an alternative model for TFS4 action could be simple direct competition for substrate NTP or
DNA, yet the authors do not mention this possibility let alone exclude it experimentally. At a minimum,
the alternative model should be described. It would be desirable to also include simple experiments to
test the direct competition models. Do low concentrations of TFS4 raise the apparent K_{ntp} for
substrates during elongation? Do changes in NTP and TFS4 concentration give competitive effects?

This is a fair point also raised by reviewer-1. We have followed up on it by carrying out additional
experiments to shed light on the mechanism of catalytic inhibition. In order to address the possibility of
a competition between TFS4- and substrate NTP binding we have carried out abortive initiation assays
using different NTP concentrations at a range of TFS4 concentrations. Our results indeed suggest that
the catalytic inhibition has a component characteristic of a *competitive inhibition*, and in molecular
terms likely reflects that TFS4 binding interferes with NTP substrate binding, or interferes with the
access of NTP substrates to their binding sites. This is in perfect agreement with the proposed binding
site of TFS4 in the NTP entry pore, and with the binding site of TFIIS in the RNAPII-TFIIS structure and
the bindings site of the bacterial transcript cleavage Gre factors, and regulators including Gfh1 and
DksA. The new results are incorporated into figure 3 and the body text of the manuscript on page 9:
*'To further elaborate on the underlying mechanism of inhibition we tested whether the TFS4 inhibition*
*varied with NTP concentrations. We carried out abortive initiation assays at a range of NTP substrate-*
*and TFS4 concentrations, and the results are illustrated as double reciprocal plot in Figure 3g (primary*
*data in supplementary Fig. 6a and b). The results show that TFS4 acts as competitive inhibitor*
*interfering with NTP binding. At 250 nM TFS4 (5-fold excess over RNAP) the apparent K_m for NTP is*
*increased ~30-fold from $27 \pm 3 \mu M$ to $870 \pm 81 \mu M$.*

Would higher concentrations of DNA lessen the inhibitory effect of TFS4 on EC dissociation?

We did carry out the template competition experiments suggested above, but found that increased
DNA concentrations did not alter the inhibition by TFS4. However, since this is negative evidence and
does not contribute significantly to the work presented in this paper we did not include it in the revised
manuscript.

2. The effects of virus infection on TFS4 expression are interesting and consistent with the authors
descriptions, but the results stop short of directly defining the role of TFS4 in phage infection. Is there
some reason TFS4 expression can't be blocked in *Sulfolobus*? (eg, a knockout mutant or some sort of

antisense RNA interference expt?). I don't know the system well enough to know how hard this would
be, but it would be very interesting to know if the virus infection is altered in a TFS4 knockout. If that's
a doable experiment, then I'd recommend trying to include it. If it's not a doable experiment, then I'd
recommend setting up the last section of the results by explaining that it would be the ideal
experiment but is not feasible in Sulfolobus and therefore an alternative approach was used (to help
readers understand why the approach used was selected).

As pointed out in the response to reviewer-1, the deleting TFS4 in *S. solfataricus* P2 is experimentally
not feasible. We have followed reviewer-2's advice and included the following section in the discussion
section of the revised manuscript:

*'Because infection by STIV strongly induces TFS4 expression, we propose that TFS4 is part of a novel*
*host response mechanism in archaea to combat viral infection. However, whether this response is*
*specific to STIV or represents a more general host response to viral infection is unknown. Future*
*experiments including mutagenesis studies of TFS4 in a crenarchaeal species that serves as host for*
*virus infection and is amenable to genetic intervention will play an important role in elucidating the*
*biological role of global transcription inhibition by TFS4.'*

3. Can the authors say anything about the relative strengths of TFS1 and TFS4 binding. Do they
compete with roughly the same affinity or does one bind much tighter than the other. It would be
particularly interesting if TFS4 bound much tighter than TFS1 because TFS1 must bind weakly
enough to allow NTP binding and tighter binding of TFS4 would make it an effective block to NTP
entry. The authors perform competition experiments so this seems like an entirely reasonable question
to address in this manuscript. The data may already exist or a simple extension of experiments already
performed would address this quite interesting question.

See above, our attempts to characterize the accurate binding affinities of TFS1 and -4 to RNAP were
not successful, while it was possible to determine the *relative* binding affinities in competition
experiments using EMSAs and transcription assays. Firstly, figure 3b shows that even at a 10-fold
excess of TFS1 over TFS4 PIC formation is not yet fully restored, which suggests that the *affinity of*
*TFS4 to the PIC is higher than TFS1*. Secondly, we carried out competition experiments using the
elongation scaffold assays shown in supplementary figure 6. TFS4 at 100nM is able to inhibit RNAP
efficiently, while TFS1 is able to counteract 50% of this inhibition at a concentration of 230 nM, in other
words in ~2.3 fold excess, which indicates that the affinity of TFS4 is higher to the TEC compared to
TFS1.

We have included the following sentence in the manuscript on page 8:
*'Since a 10-fold excess of TFS1 over TFS4 barely restores the PIC signal in the EMSA the relative*
*affinity of TFS4 for the DNA-TBP-TFB-RNAP complex is higher than that of TFS1 (Fig. 3b).'*
And on page 11:
*'Considering that the RNAP binding sites of TFS1 and -4 likely are identical or at least overlap, we*
*wanted to test whether TFS1 was able to counteract TFS4 inhibition in transcription elongation assays.*
*Under our experimental conditions a two-fold excess of TFS1 over TFS4 is required to achieve 50%*
*relief of inhibition, which indicates that TFS4 has a higher relative affinity for RNAP in the TEC*
*compared to TFS1 (Supplementary Fig. 8a and b).'*

Minor points

1. Abstract, third line and several other places in the manuscript. "While" is best reserved to mean
"contemporaneously" - "Although" is a better word choice here.

We introduced the suggested change.

2. Abstract line 8. Although I agree an "allosteric" model is a reasonable explanation, since the
experiments do not provide unambiguous discrimination against a direct competition model it might
be safest to be more conservative in the abstract.

Our new data demonstrate that the catalytic inhibition is, at least in part, competitive in terms of NTP
binding. Our direct binding EMSA experiments show clearly that TFS4 counteracts transcription
complex stability of PIC and TEC. Since TFS4 enters the RNAP through the NTP channel and does not
compete with DNA template binding, the destabilising effect is *very likely* to be allosteric, and in our
opinion it is well justified to keep the term 'allosteric' in the abstract. If the editor feels strongly about
this point, we will naturally concur and moderate our language.

3. p3 li5. The "secondary channel" is generally accepted nomenclature for bacterial RNAP (and
perhaps for archaeal RNAP – I'm uncertain in that case), but for eukaryotic RNAPII the terms "pore"
and "funnel" are the generally accepted nomenclature for the outer and inner parts of the secondary
channel. It might help target a broader audience to explain both sets of terms here.

The field should strive towards a unified nomenclature of RNAP motifs and domains. Yet, there is no
unambiguous nomenclature adapted for archaeal RNAP. For improved clarity, we settled on the use of
the term *secondary channel*. A sentence was added in Introduction on page 3:

*'The shape of the universally conserved RNAP core resembles a crab claw with a DNA binding*
*channel facilitating interactions with the DNA template, and a secondary channel, also called the NTP*
*entry channel or funnel and pore motifs, in eukaryotic RNAPs².'*

4. p3 li9 should be a comma after phase

We introduced the suggested change.

5. p3 li11 It is more accurate from a cellular perspective to say that DNA moves backwards through
RNAP than that RNAP moves backwards on DNA since the DNA is generally more mobile than the
RNAP. Either frame of reference is correct, however.

This is a very interesting almost philosophical consideration, did the mountain go to the prophet or vice
versa? We did not change the sentence since reviewer-2 concedes that both frames of reference are
factually correct.

6. p3 third line from bottom "urgent" implies a need to act quickly. Perhaps "strong" would be a
better choice.

We introduced the suggested change.

7. p4 li17 Generally, cleavage factors are thought to stabilize binding of the second Mg rather than
alter the position of binding and it might be better to substitute stabilize for allow here.

We introduced the term 'stabilise'.

8. p4 li22 The first sentence of this para is a little awkward. Perhaps something like "Here we describe
the discovery of a fascinating multiplication of tfs-like genes in archaea and present the
characterization of two of them, TFS1 and TFS4, that display opposite stimulatory and inhibitory effects
on transcription.

We introduced the suggested change.

9. p4 third line from the bottom. This itemized list is not parallel; (i) destabilizes, (ii) inhibits, (iii) no verb,
(iv) no verb. Rephrase to create a parallel construction.

We introduced the suggested change, which now reads:

*'TFS4 inhibits (i) PIC formation, (ii) abortive initiation, (iii) promoter-directed transcription, as well as (iv)*
*transcription elongation.'*

10. p5 li3 typo "tree" should be "three"

We introduced the suggested change.

11. p6 li16 Is it possible that His residues substitute for the missing Cys residues in the ZR domains?

Histidine residue have been shown to be able to substitute for cysteine residue in Zinc-ribbon (ZR)
domains, however TFS3 does not encode any suitably proximal histidine residues making the
presence of ZR domains unlikely.

12. p7 li7 Fig. 2a does not show the template referred to in this citation. It would be very helpful to
readers to depict the template structure/sequence in key segments and a brief experimental
schematic.

We introduced the suggested change and included the sequences of DNA templates used in this study
in supplementary figure 12.

13. p8 li15 It is unclear to me why the inhibition of PIC formation necessarily means an allosteric effect.
Couldn't parts of TFS4 be located where they directly compete with DNA?

Without a high resolution structure of the TFS4-RNAP complexes a steric occlusion mechanism
between TFS4 and the template DNA cannot be ruled out. However, all our results suggest that TFS4
binds in the secondary channel of RNAP, a position which is not likely compatible with an DNA
template competition mechanism.

14. p9 li11 The difference in salt concentrations for high vs low is not particularly large. Is Cl even the
relevant counterion for in vivo. For many organisms Cl is excluded and can compete for
proteinphosphate ionic interactions.

The accurate characterization of the intracellular ionic strength and ionic composition is very
challenging. To the best of our knowledge no data are available regarding the intracellular salt
concentrations in any *Sulfolobus* species. However, based on the growth medium composition, the
external chloride concentration is below 1 mM and the overall ionic strength is below 15 mM. Our *in*
*vitro* experiments are all carried out using *stringent conditions*, and the binding and reaction buffers are
optimized to minimize background interactions and to obtain optimal specificity; both salt
concentrations (150 and 250 mM) are likely above the *predicted* intracellular salt concentrations. The
fact remains that TFE alpha/beta function is sensitive to the relatively small differences in chloride
concentrations.

15. p10 li4 Wasn't an effect of TFS4 on inhibition of elongation shown in Fig. 2?

Figure 2 shows that TFS4 cannot reactivate arrested elongation complexes, while figure 4 shows that
TFS4 inhibits transcription elongation complexes.

16. p11 last line. This is a point where it's particularly unclear why the simple steric inhibition of NTP
binding isn't considered as an explanation for inhibitory effects on elongation.

See above, our kinetic experiments show that TFS4 inhibition indeed is dependent on the NTP
concentration, and we are discussing this new finding in the revised manuscript.

17. p12 li14 use of two "which" in the independent clause is confusing.

We altered the sentence to the following:

'*As a negative control, we monitored the protein levels of the RNAP subunit Rpo7 and the chromatin*
*protein Alba, which remained unchanged during the infection time course. This suggests that the*
*increase in TFS4 levels was specific to TFS4 and not due to a general decrepitude as a result of virus*
*infection (Fig. 6c).'*

18. p15 li7 There also are data on a Gre mutant in which substitution of one or two of the acidic

residues turns it into an inhibitor. I apologize for not looking up the citation, but I think it's work from
Arkady Mustaev, and would be worth citing here.

We have introduced the suggested reference (Laptenko et al., EMBO J, 2003), which describes that
the alanine substitution variants of GreA's acidic residues turn the factor into a inhibitor.

19. p15 li11 Again here, it's unclear why the direct competition for NTP binding isn't considered.

See above, our kinetic experiments show that TFS4 inhibition indeed is dependent on the NTP
concentration, and we are discussing this new finding in the revised manuscript.

20. Fig. 4 Why is the formation of ECs so inefficient? What percentages of DNA and RNAP are formed
into EC? Since it's hard to see appearance of DNA released from EC upon TFS4 binding, the
conclusion that TFS4 disrupts EC is not particularly well supported by the experiment. Isn't it possible
that DNA remains bound and TFS4 binding just causes the EC to smear out during electrophoresis?

Following both reviewer-1's and -2's recommendations we have repeated the TEC EMSA assays. The
new optimized binding experiments show very clearly (i) a superhift of the RNAP-DNA-RNA upon the
addition of TFS1 reflecting RNAP-DNA-RNA-TFS1 complex formation, and (ii) that the addition of
TFS4 disrupts the RNAP-DNA-RNA complex. For direct comparison we have used the same
factor:RNAP ratios for TFS1 and -4.

21. Fig. 5d and 5e Is the x-axis correctly labeled? It seems like it should be TFS4/RNAP ratio. Also,
maybe don't use gray for the triple A mutant – it's hard to distinguish from black on the figure.

We introduced the suggested change.

Reviewers' Comments:

Reviewer #1 (Remarks to the Author):

Authors have made corrections in the text and Figures and addressed all my questions and concerns. Authors also added new important data on the nature of TFS4 inhibitory activity which I think significantly improves the paper. It can now be published without further revision.

Reviewer #2 (Remarks to the Author):

Fouqueau et al have done an admirable job of revising their manuscript in response to suggestions from the reviewers, and the revised manuscript is largely suitable for publication.

In finalizing the manuscript, the authors may wish to consider slightly modifying and clarifying their explanation for effects of TFS4 on elongation. Reviewer 1 was concerned about the lack of shorter product accumulation in fig 4a and whether this was inconsistent with inhibition of catalysis. The competitive inhibition assay conducted by the authors supports their conclusion, but they may wish to additionally point out that inhibition of catalysis will not result in accumulation of shorter products if the NTP binding constant for the first round of nucleotide addition during initiation is much higher than the average NTP binding constant during elongation. This disparity in NTP binding affinities is typical for multisubunit RNA polymerases because the first round of nucleotide addition requires two NTPs instead of one, and the position of the template strand is not stabilized by an RNA-DNA hybrid. Additionally, the authors may wish to consider whether describing the appearance of shorter transcripts in fig 4b using the TFS4 mutant reflects “reduced transcription processivity”. First, as processivity is loosely defined in the contemporary transcription field, it is often used to mean that RNAP has a greater or lesser tendency to release the DNA template. The appearance of the shorter transcripts could reflect release of the template but could also just reflect a position at which the TFS4 mutant is unusually good at inhibiting nucleotide addition, thus slowing the polymerase without releasing the template. More information would be needed to assess template release. Second, in polymer enzymology, processivity is measure of the ability of a chain to be extended by a processively bound enzyme rather than by a new enzyme molecule. Thus reduced processivity would not necessarily lead to an accumulation of shorter products just a change in how they are elongated. Because of this confusion, processivity may be a term best avoided in the transcription field.

These are minor points that the authors can consider, but overall the manuscript is in good shape for publication.

Detailed response to reviewer's comments

Reviewer #1 (Remarks to the Author):

Authors have made corrections in the text and Figures and addressed all my questions and concerns. Authors also added new important data on the nature of TFS₄ inhibitory activity which I think significantly improves the paper. It can now be published without further revision.

Reviewer #2 (Remarks to the Author):

Fouqueau et al have done an admirable job of revising their manuscript in response to suggestions from the reviewers, and the revised manuscript is largely suitable for publication.

In finalizing the manuscript, the authors may wish to consider slightly modifying and clarifying their explanation for effects of TFS₄ on elongation. Reviewer 1 was concerned about the lack of shorter product accumulation in fig 4a and whether this was inconsistent with inhibition of catalysis. The competitive inhibition assay conducted by the authors supports their conclusion, but they may wish to additionally point out that inhibition of catalysis will not result in accumulation of shorter products if the NTP binding constant for the first round of nucleotide addition during initiation is much higher than the average NTP binding constant during elongation. This disparity in NTP binding affinities is typical for multisubunit RNA polymerases because the first round of nucleotide addition requires two NTPs instead of one, and the position of the template strand is not stabilized by an RNA-DNA hybrid.

These are valid considerations, and we have incorporated this statement into the revised version of the manuscript:

'The inhibition of catalysis would not result in accumulation of shorter products if the NTP binding constant for the first round of nucleotide addition during initiation is much higher than the average NTP binding constant during elongation. This disparity in NTP binding affinities is typical for multisubunit RNA polymerases because the first round of nucleotide addition requires two NTPs instead of one, and the position of the template strand is not stabilized by an RNA-DNA hybrid.'

Additionally, the authors may wish to consider whether describing the appearance of shorter transcripts in fig 4b using the TFS₄ mutant reflects "reduced transcription processivity".

This comment is likely based on a misunderstanding, **figure 4b** shows indeed an accumulation of partial products but this is caused by the TFS₁-AA variant, and not TFS₄ at all, and hence the considerations below are not relevant.

Detailed response to reviewer's comments

First, as processivity is loosely defined in the contemporary transcription field, it is often used to mean that RNAP has a greater or lesser tendency to release the DNA template. The appearance of the shorter transcripts could reflect release of the template but could also just reflect a position at which the TFS₄ mutant is unusually good at inhibiting nucleotide addition, thus slowing the polymerase without releasing the template. More information would be needed to assess template release. Second, in polymer enzymology, processivity is measure of the ability of a chain to be extended by a processively bound enzyme rather than by a new enzyme molecule. Thus, reduced processivity would not necessarily lead to an accumulation of shorter products just a change in how they are elongated. Because of this confusion, processivity may be a term best avoided in the transcription field.

We do appreciate the complications arising from using the term processivity and have removed it from the original statement characterising the TFS₁-AA variant, which now reads:

'Interestingly the TFS₁-AA mutant leads to the accumulation of partial 45/46 nt transcripts (Fig. 4b)' instead of 'Interestingly the TFS₁-AA mutant reduced transcription processivity, which is reflected in the accumulation of partial 45/46 nt transcripts (Fig. 4b).'

These are minor points that the authors can consider, but overall the manuscript is in good shape for publication.